# Persistent firing in LEC III neurons is differentially modulated by learning and aging

**Carmen Lin\*, Venus N Sherathiya, M Matthew Oh†, John F Disterhoft†\***

Department of Physiology, Feinberg School of Medicine, Northwestern University, Chicago, United States

**Abstract** Whether and how persistent firing in lateral entorhinal cortex layer III (LEC III) supports temporal associative learning is still unknown. In this study, persistent firing was evoked in vitro from LEC III neurons from young and aged rats that were behaviorally naive or trained on trace eyeblink conditioning. Persistent firing ability from neurons from behaviorally naive aged rats was lower compared to neurons from young rats. Neurons from learning impaired aged animals also exhibited reduced persistent firing capacity, which may contribute to aging-related learning impairments. Successful acquisition of the trace eyeblink task, however, increased persistent firing ability in both young and aged rats. These changes in persistent firing ability are due to changes to the afterdepolarization, which may in turn be modulated by the postburst afterhyperpolarization. Together, these data indicate that successful learning increases persistent firing ability and decreases in persistent firing ability contribute to learning impairments in aging.

**\*For correspondence:**
carmen-lin@u.northwestern.edu
(CL);
jdisterhoft@northwestern.edu
(JFD)

†These authors contributed
equally to this work

**Competing interests:** The
authors declare that no
competing interests exist.

**Reviewing editor:** Lisa
Giocomo, Stanford School of
Medicine, United States

## Introduction

Persistent firing is the ability of a neuron to continuously fire action potentials even after the termination of a triggering stimulus. Temporal associative learning and other working memory-dependent tasks are thought to rely on persistent firing because this mechanism may serve to bridge the temporal gap between task-relevant stimulus presentation and the execution of a learned response (*Zylberberg and Strowbridge, 2017*; *Tsao et al., 2018*). Evidence that persistent firing may support temporal associative learning comes from an in vivo study of prefrontal neurons, which showed elevated activity during the delay period of a working memory task (*Fuster, 1972*).

One region that has been shown to be important for temporal associative learning is the lateral entorhinal cortex (LEC) (*Morrissey et al., 2012*; *Wilson et al., 2013*). In vitro, persistent firing in the LEC is dependent upon cholinergic activation (*Tahvildari et al., 2007*; *Tahvildari et al., 2008*). Application of a cholinergic antagonist into the entorhinal cortex impairs the acquisition of temporal associative learning tasks such as trace fear and trace eyeblink conditioning (*Esclassan et al., 2009*; *Tanninen et al., 2015*). Within the LEC, layers III and V pyramidal neurons have been shown to be capable of persistent firing (*Tahvildari et al., 2007*; *Tahvildari et al., 2008*). Notably, layer III pyramidal neurons integrate and provide object information directly to the CA1 region of the hippocampus, forming the temporoammonic pathway (*van Groen et al., 2003*; *Hargreaves et al., 2005*; *Deshmukh and Knierim, 2011*). Thus, persistent firing by layer III LEC pyramidal neurons is a prime candidate to support temporal associative learning. However, how this may occur is yet to be determined.

Along with its role in hippocampus-dependent learning, the LEC is also highly susceptible to aging-related changes. The LEC is the initial site of Alzheimer's-related tau pathology in cortical regions (*Moryś et al., 1994*; *Gómez-Isla et al., 1996*; *Stranahan and Mattson, 2010*; *Khan et al., 2014*). Several studies have also determined that changes to LEC structure and activity in normal

aging is associated with learning and other cognitive impairments (*Rodrigue and Raz, 2004*; *Yassa et al., 2010*). For example, LEC hypoactivity has been shown to be associated with impairments in pattern separation in aging humans (*Reagh et al., 2018*). As a result of its susceptibility to aging, there may be alterations to persistent firing ability in LEC III that may contribute to the aging-related learning impairments we and others have observed (*Thompson et al., 1996*; *Knuttinen et al., 2001*; *Tombaugh et al., 2005*; *Disterhoft and Oh, 2006*; *Galvez et al., 2011*).

The purpose of the present study was to examine whether persistent firing properties of LEC III pyramidal neurons are altered as a result of learning and/or aging. Persistent firing was evoked from neurons of young adult and aged rats that were behaviorally naïve or trained on trace eyeblink conditioning, a temporal associative learning task. Measurements of the postburst afterhyperpolarization (AHP) and afterdepolarization (ADP) were also made to determine the mechanisms underlying these changes. The results of this study are a first step in understanding how persistent firing supports trace eyeblink conditioning and how this function is altered with aging.

## Results

### Persistent firing probability is decreased with aging

To determine whether aging decreased persistent firing ability in LEC III neurons, persistent firing was evoked in neurons from behaviorally naive young adult (Young Naive – YN) and aged (Aged Naive – AN) male rats based on protocols previously described (*Tahvildari et al., 2007*; *Tahvildari et al., 2008*; *Yoshida et al., 2008*; *Batallán-Burrowes and Chapman, 2018*). In the presence of 10 µM carbachol, the neuron's membrane potential was held at 2 mV more hyperpolarized than spontaneous firing threshold while the cell was injected with a 2 s long depolarizing current step injection of 100 pA, 150 pA, and 200 pA (*Figure 1A*). Each current injection was applied three times to obtain an average firing probability. An incidence of persistent firing was considered if firing activity occurred within 10 s after the end of the current injection. Neurons from AN animals had a lower probability of persistent firing compared to neurons from YN rats, which were able to fire close to 100%, when injected with a 100 pA and a 150 pA stimulus (*Figure 1B*; *Table 1*).

To ensure that differences in probability were not due to differences in the number of action potentials evoked during the current injection, the neurons were injected with a 2 s long train of 2 ms, 2 nA current pulses at 20 Hz to elicit a total of 40 action potentials to evoke persistent firing (*Figure 1C*). Previous studies determined that a 2 s long depolarizing current step that elicited firing at 20–30 Hz was sufficient to evoke persistent firing (*Tahvildari et al., 2007*). When persistent firing was evoked in neurons from YN and AN rats with this protocol, neurons from AN rats again had a lower probability of firing compared to neurons from YN rats, which were able to fire at nearly 100% (*Figure 1D*; *Table 2*). We also evoked persistent firing using a 250 ms long train of 2 ms, 2 nA current pulses at 20 Hz and held the neuron's membrane potential at 2 mV and 5 mV more hyperpolarized than spontaneous firing threshold (*Figure 1E,G*). This protocol was used to be more reflective of the trace eyeblink conditioning task we used (a 250 ms long CS). We also wanted to determine whether persistent firing could be evoked in vitro under more physiological conditions involving shorter stimulation time and naturally occurring spontaneous Up-States (*Hahn et al., 2012*). Persistent firing was able to be evoked in neurons from YN and AN rats using a 250 ms long current injection with the membrane potentials held at 2 mV and 5 mV more hyperpolarized than spontaneous firing threshold. With the membrane potential at 2 mV more hyperpolarized, the firing probability of neurons from YN animals was 100%, although it was again lower in neurons from AN (*Figure 1F*; *Table 2*). When the neuronal membrane potential was held at 5 mV below spontaneous firing threshold, the probability of firing was lower for neurons in both groups overall, and there was no statistical difference in probability between the young and aged groups (*Figure 1H*; *Table 2*).

Together, these data indicate that neurons from behaviorally naive aged animals overall have a lower probability of persistent firing compared to neurons from young animals, which were nearly always able to fire persistently. The decreased probability of firing from aged animals may contribute to the aging-related learning impairments in trace eyeblink conditioning that we have previously observed (*Thompson et al., 1996*; *Knuttinen et al., 2001*; *Disterhoft and Oh, 2006*; *Galvez et al., 2011*).

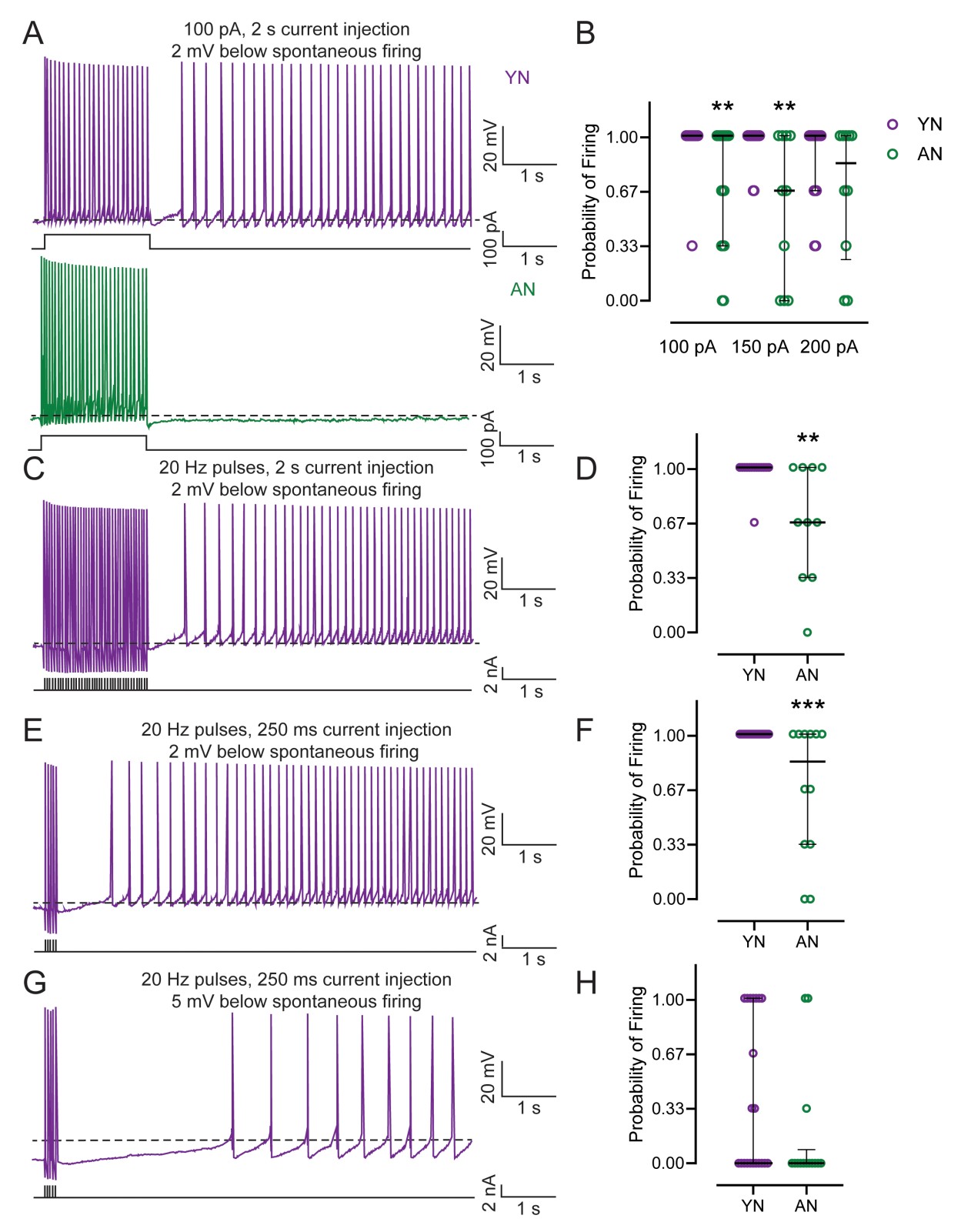

**Figure 1.** Persistent firing probability is decreased in neurons from behaviorally naïve aged animals. (A) Persistent firing example traces from neurons from behaviorally naive young (Young Naive – YN; *top*, purple) and aged (Aged Naive – AN; *bottom*, dark green) animals. Persistent firing was evoked with a 100 pA 2 s long current injection while the neuronal membrane potential was held at 2 mV more hyperpolarized than spontaneous firing. Dotted lines indicate spontaneous firing threshold. Black lines underneath persistent firing activity indicate the current injections used. (B) Neurons from AN

*Figure 1 continued on next page*

*Figure 1 continued*

animals have a lower probability of firing relative to neurons from YN animals when neurons are injected with a 100 pA (AN, n = 18 neurons; YN, n = 25), 150 pA (AN, n = 10; YN, n = 18), but not a 200 pA training stimulus (AN, n = 10; YN, n = 16). (Mann-Whitney, 100 pA U = 134.5, **p=0.0016; 150 pA U = 42, **p=0.0029; 200 pA U = 60.50, p=0.2458). Error bars: median and quartiles (75%–25%). (C) Persistent firing example trace from YN neuron evoked with 2 s long 20 Hz train of current pulses. Neuronal membrane potential held at 2 mV more hyperpolarized than spontaneous firing threshold. Dotted lines indicate spontaneous firing threshold. Black lines underneath persistent firing activity indicate the current used. (D) Neurons from AN animals (n = 10 neurons) have a lower probability of firing relative to neurons from YN animals when persistent firing is evoked with 2 s long 20 Hz train of current pulses (n = 14). (Mann-Whitney, U = 31.50, **p=0.0052). Error bars: median and quartiles (75%–25%). (E) Persistent firing example trace from YN neuron evoked with 250 ms long 20 Hz train of current pulses. Neuronal membrane potential held at 2 mV more hyperpolarized than spontaneous firing threshold. Dotted lines indicate spontaneous firing threshold. Black lines underneath persistent firing activity indicate the current used. (F) Neurons from AN animals (n = 12 neurons) have a lower probability of firing relative to YN (n = 22) when persistent firing is evoked with a 250 ms long 20 Hz train of current pulses and the neuronal membrane potential is 2 mV more hyperpolarized than spontaneous firing threshold. (Mann-Whitney, U = 66, ***p=0.0007). Error bars: median and quartiles (75%–25%) (G) Persistent firing example trace from YN neuron evoked with 250 ms long 20 Hz train of current pulses. Neuronal membrane potential held at 5 mV more hyperpolarized than spontaneous firing threshold. Dotted lines indicate spontaneous firing threshold. Black lines underneath persistent firing activity indicate the current used. (H) No difference in probability between neurons from AN (n = 14 neurons) and YN (n = 21) animals when persistent firing is evoked with a 250 ms long 20 Hz train of current pulses and the membrane potential is 5 mV more hyperpolarized than spontaneous firing threshold. (Mann-Whitney, U = 107.5, p=0.1304). Error bars: median and quartiles (75%–25%). See *Tables 1* and *2*.

## Successful learning enhances persistent firing probability, and learning impairments are associated with lower firing probability

We next examined the relationship between learning and persistent firing ability. Our objective was to determine whether the aging-related decrease in persistent firing probability would be reflected amongst learning impaired aged animals. We also wanted to determine if successful learning would increase persistent firing probability. To that end, we examined persistent firing in LEC III pyramidal neurons from aged and young adult rats that had been trained on trace eyeblink conditioning.

During conditioning, a 250 ms long auditory tone conditioned stimulus (CS) was paired with a 100 ms long electrical shock to the periorbital region, which served as the unconditioned stimulus (US). The CS and the US were separated by a 500 ms long stimulus-free trace interval (*Figure 2A*). The aged rats were separated into aged impaired (AI) or aged unimpaired (AU) groups based upon whether they achieved a 60% conditioned response (CR) learning criterion. 60% CRs were chosen based on the separation of behavior amongst aged animals on the last session of trace eyeblink conditioning (*Figure 2B*). The young adult rats were either trained on trace eyeblink conditioning (Young Conditioned – YC) or pseudoconditioned (Young Pseudoconditioned – YP) as a control.

Our learning curves show a steady increase across the training sessions in % CRs from the groups that were trained on the trace eyeblink conditioning paradigm and were able to acquire it (YC and AU) (RM ANOVA, session, $F_{4.138, 302.1}$ = 112.6, p<0.0001; group x session, $F_{15, 365}$ = 14.97, p<0.0001). YP and AI animals did not exhibit a discernable increase in % CR. YC rats showed the fastest rate of increase in % CR, exhibiting more CRs within the first session than YP, AI, or even AU. Although AU were slower than YC rats in learning trace eyeblink conditioning, they were

**Table 1.** Persistent firing probability evoked with a 100 pA, 150 pA, and 200 pA training stimulus with statistical differences between YN and AN.
Related to *Figure 1B*.

| Group | n | 100 pA (%) | p-value | n | 150 pA (%) | p-value | n | 200 pA (%) | p-Value |
|-------|---|-----------|---------|---|-----------|---------|---|-----------|---------|
| YN | 25 | 97.3 ± 2.7 | | 18 | 96.3 ± 2.5 | | 16 | 85.4 ± 6.1 | |
| AN | 18 | 72.2 ± 8.6 | vs. YN, 0.0016** | 10 | 56.7 ± 14.1 | vs. YN, 0.0029** | 10 | 66.7 ± 13.2 | vs. YN, 0.2458 |

| | 100 pA | | | 150 pA | | | 200 pA | | |
|-------|---------|--------|---------|------|--------|------|------|--------|------|
| Group | Q1 (25%) | Median | Q3 (75%) | Q1 | Median | Q3 | Q1 | Median | Q3 |
| YN | 1.00 | 1.00 | 1.00 | 1.00 | 1.00 | 1.00 | 0.67 | 1.00 | 1.00 |
| AN | 0.33 | 1.00 | 1.00 | 0.00 | 0.67 | 1.00 | 0.25 | 0.83 | 1.00 |

*n*, number of cells in group; *p*-value, Mann-Whitney.

**Table 2.** Persistent firing probability evoked with a 20 Hz current pulses at various stimulus lengths and membrane holding potentials with statistical differences between YN and AN.
Related to *Figure 1D, F, H*.

| Group | n | 20 Hz, 2 s, 2 mV below (%) | p-Value | n | 20 Hz, 250 ms, 2 mV below (%) | p-Value | n | 20 Hz, 250 ms, 5 mV below (%) | p-Value |
|---|---|---|---|---|---|---|---|---|---|
| YN | 14 | 97.6 ± 2.4 | | 22 | 100.0 ± 0.0 | | 21 | 39.7 ± 10.2 | |
| AN | 10 | 66.7 ± 11.1 | vs. YN, 0.0052** | 12 | 66.7 ± 11.6 | vs. YN, 0.0007** | 14 | 16.7 ± 9.7 | vs. YN, 0.1304 |

| | 20 Hz, 2s, 2 mV below | | | 20 Hz, 250 ms, 2 mV below | | | 20 Hz, 250 ms, 5 mV below | | |
|---|---|---|---|---|---|---|---|---|---|
| Group | Q1 (25%) | Median | Q3 (75%) | Q1 | Median | Q3 | Q1 | Median | Q3 |
| YN | 1.00 | 1.00 | 1.00 | 1.00 | 1.00 | 1.00 | 0.00 | 0.00 | 1.00 |
| AN | 0.33 | 0.67 | 1.00 | 0.33 | 0.83 | 1.00 | 0.00 | 0.00 | 0.08 |

*n*, number of cells in group; *p*-value, Mann-Whitney.

indiscernible from YC in terms of % CR by the end of training (*Figure 2B*; *Table 3*). Persistent firing and other intrinsic neuronal properties were measured the day after the last training session.

We evoked persistent firing using a 250 ms long 20 Hz train of 2 ms, 2 nA current pulses. The neuronal membrane potential was held at both 2 mV and 5 mV more hyperpolarized from the neuron's spontaneous firing threshold. As neurons from YN animals were able to persistently fire at 100% probability when the neuronal membrane potential was held at 2 mV more hyperpolarized than spontaneous firing threshold, there was a ceiling effect on persistent firing probability from YC and YP animals at this membrane potential. Neurons from AU animals were also able to reach the same ceiling effect that we observed in young animals. Neurons from AI, however, had a lower probability of firing compared to AU, YC, and YP (Kruskal-Wallis, H = 15.27, p=0.0016) (*Figure 2C*; *Table 4*).

Although LEC III pyramidal neurons from YN and AN animals could fire when the membrane potential was held at 5 mV below spontaneous firing threshold, the probability of firing was significantly reduced (*Figure 1H*). Successful acquisition of trace eyeblink conditioning, however, significantly increased the probability of firing at this membrane potential from both young adult and aged animals (Kruskal-Wallis, H = 30.86, p<0.0001). In YC, neurons reached 100% firing, while neurons from YP had a lower probability, firing at levels similar to those of behaviorally naïve young animals (*Figure 2—figure supplement 1*; *Table 5*). Although AU did not achieve the same ceiling level of firing as YC, they were statistically comparable to YC as well as to YP (*Figure 2D*; *Table 4*) and had increased firing ability relative to neurons from AI and AN animals (*Figure 2D*; *Table 4*; *Figure 2—figure supplement 1*; *Table 5*). AI had the lowest probability of firing relative to the other groups (*Figure 2D*; *Table 4*). The significance of learning and learning impairments on persistent firing ability is demonstrated in *Figure 3*, which shows the persistent firing activity of each neuron represented in *Figure 2D*. Neurons from YC are the most active and have the highest spike count per

**Table 3.** Learning curves with statistical differences between YC, YP, AU, and AI.
Related to *Figure 2B*.

| Group | n | Session 1 (% Late CR's) | p-Value | Session 5 (% Late CR's) | p-Value |
|---|---|---|---|---|---|
| YC | 21 | 34.5 ± 4.9 | vs. YP, 0.0035** | 74.3 ± 3.2 | vs. YP, <0.0001*** |
| YP | 19 | 13.6 ± 2.6 | | 18.7 ± 3.6 | |
| AU | 21 | 17.7 ± 2.9 | vs. YC, 0.0266* | 69.9 ± 3.2 | vs. YC, 0.7789 |
| | | | vs. YP, 0.7199 | | vs. YP, <0.0001*** |
| AI | 16 | 11.9 ± 2.1 | vs. AU, 0.3710 | 29.1 ± 4.1 | vs. AU, <0.0001*** |
| | | | vs. YC, 0.0011** | | vs. YC, <0.0001*** |
| | | | vs. YP, 0.9518 | | vs. YP, 0.2567 |

*n*, number of rats in group; *p*-value, Tukey's multiple comparisons test.

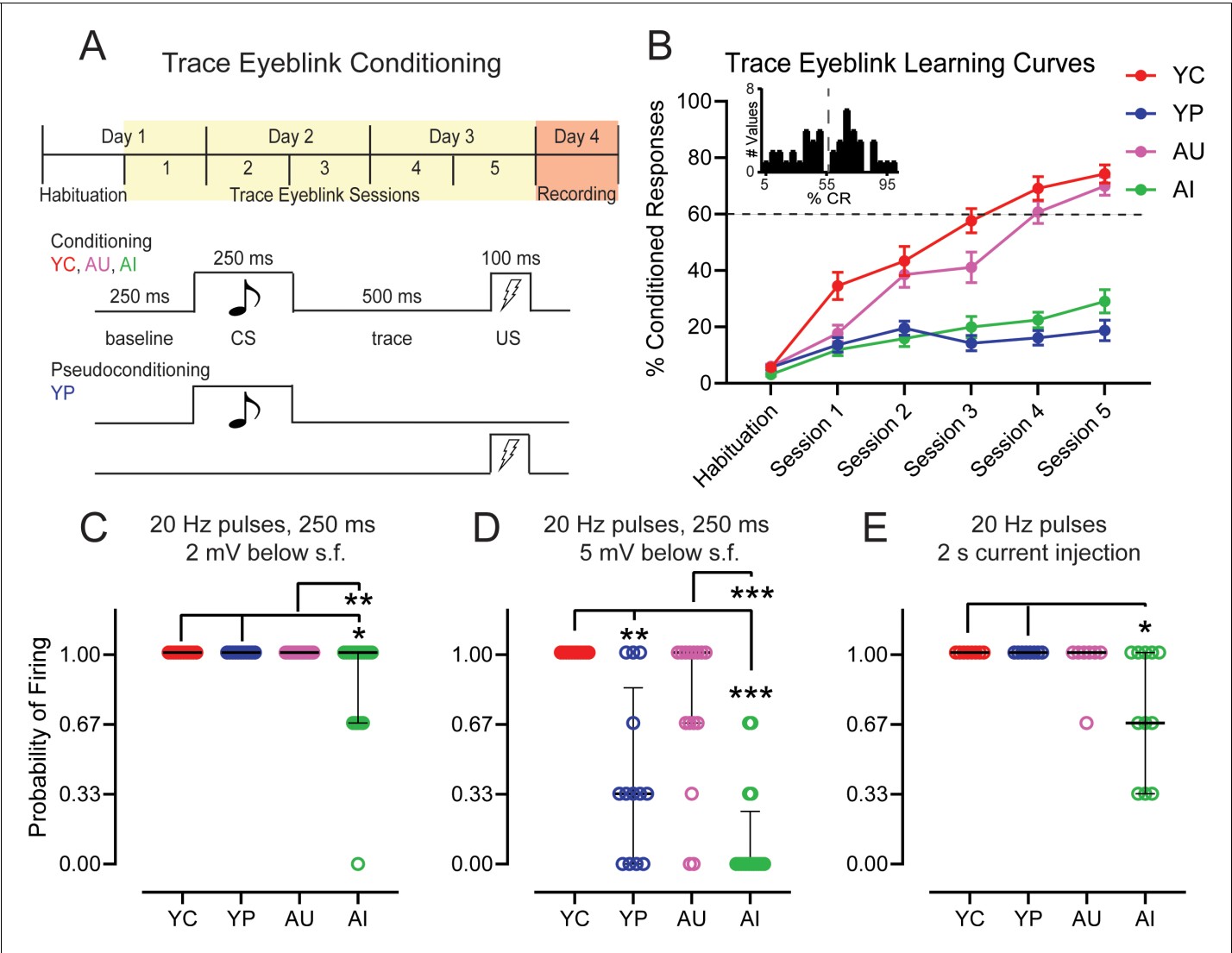

**Figure 2.** Neurons from aged impaired animals have a decreased persistent firing probability, and successful learning increases persistent firing probability in neurons from both young adult and aged animals. (**A**) Trace Eyeblink Conditioning Paradigm. *Top,* Young adult and aged rats were trained on trace eyeblink conditioning over the course of 3 days. The first session of Day 1 was a habituation session to the training apparatus. Following habituation, the rats were given five sessions of eyeblink conditioning. Biophysical recordings were performed 24 hrs after the last session of eyeblink conditioning. *Bottom,* Conditioned and Pseudoconditioned paradigms. Aged animals were trained on the conditioning paradigm and separated into Aged Unimpaired (AU) and Aged Impaired (AI) groups depending on learning ability. Young adult animals were conditioned (Young Conditioned – YC) or pseudoconditioned (Young Pseudoconditioned – YP). In the Conditioning Paradigm, animals are presented with a 250 ms long tone Conditioned Stimulus (CS) paired with a 100 ms long electrical shock to the periorbital region Unconditioned Stimulus (US), separated with a 500 ms long stimulus-free trace period. (**B**) YC (n = 21 rats; red) and AU (n = 21; pink) animals successfully acquire trace eyeblink conditioning. Criterion for successful learning is 60% Conditioned Responses (CRs) (dotted line). AI (n = 16; lime green) animals are unable to achieve 60% CRs, while YP (n = 19; blue) animals do not receive paired CS-US stimuli. Error bars: mean ± SEM. *Inset,* frequency distribution of %CR from aged animals in Session 5 of trace eyeblink conditioning shows a separation in behavior at around 55–60% CR. (**C**) Neurons from AI animals (n = 18 neurons) have a lower probability of firing relative to neurons from YC (n = 12), YP (n = 11), and AU (n = 19) animals when persistent firing is evoked with a 250 ms long 20 Hz train of current pulses and the neuronal membrane potential is 2 mV more hyperpolarized than spontaneous firing threshold. (Dunn's, \*\*p=0.0049; AI vs. YC \*p=0.0188; AI vs. YP \*p=0.0241). Error bars: median and quartiles (75%–25%). (**D**) Learning enhances persistent firing probability in neurons from YC animals (n = 11 neurons), relative to neurons from YP animals (n = 13). Neurons from AI animals (n = 16) have a lower probability of firing, relative to neurons from YC and AU animals (n = 15). Persistent firing evoked with a 250 ms long 20 Hz train of current pulses and the membrane potential is 5 mV more hyperpolarized than spontaneous firing threshold. (Dunn's, AI vs. YC \*\*\*p<0.0001; AI vs. AU \*\*\*p=0.0008; \*\*p=0.0063). Error bars: median and quartiles (75%–25%). (**E**) Neurons from AI animals (n = 11 neurons) have a lower probability of firing relative to neurons from YC (n = 8), and YP (n = 8), but not AU (n = 7) animals when persistent firing is evoked with a 2 s long train of 20 Hz current pulses and the neuronal membrane potential is 2 mV more

*Figure 2 continued on next page*

*Figure 2 continued*

hyperpolarized than spontaneous firing threshold. (Dunn's AI vs. YC *p=0.0236; AI vs. YP *p=0.0236). Error bars: median and quartiles (75%–25%). See **Tables 3–6**.

The online version of this article includes the following figure supplement(s) for figure 2:

**Figure supplement 1.** Neurons from aged impaired animals have a decreased probability of firing, when persistent firing is evoked with a 2 s long rectangular current injection and the membrane potential is held at 2 mV below spontaneous firing threshold.

**Figure supplement 2.** Neurons from YC had increased persistent firing probability compared to YP and YN, and neurons from AU had increased persistent firing probability compared to AI and AN.

cell across time. In contrast, neurons from AI are the least active and have the lowest spike count per cell across time (**Figure 3**).

To ensure that the decreased firing probability from AI was not merely due to a shortened stimulation time, we also evoked persistent firing using a 2 s long 20 Hz train of current pulses with the membrane potential held at 2 mV more hyperpolarized than spontaneous firing threshold. When we evoked persistent firing using this protocol, AI again was the only group that did not have a ceiling or near-ceiling firing probability (Kruskal-Wallis, H = 12.00, p=0.0074) (**Figure 2E**, **Table 4**). A similar effect was also observed when the neurons were injected with a 100 pA (Kruskal-Wallis, H = 19.09, p=0.0003), 150 pA (Kruskal-Wallis, H = 14.85, p=0.0020), and 200 pA (Kruskal-Wallis, H = 10.29, p=0.0162) 2 s current step injection (**Figure 2—figure supplement 2**; **Table 6**).

## Aging decreases persistent firing rate while learning increases firing rate

Given the significant impact that both aging and learning have on persistent firing probability, our next question was whether these factors would have a similar effect on various properties of persistent firing, such as persistent firing rate and onset latency.

**Table 4.** Persistent firing probability evoked with a 20 Hz current pulses at various stimulus lengths and membrane holding potentials with statistical differences between YC, YP, AU, and AI.
Related to **Figure 2C, D, E**.

| Group | n | 20 Hz, 2 s, 2 mV below (%) | p-Value | n | 20 Hz, 250 ms, 2 mV below (%) | p-Value | n | 20 Hz, 250 ms, 5 mV below (%) | p-Value |
|---|---|---|---|---|---|---|---|---|---|
| YC | 8 | 100.0 ± 0.0 | vs. YP, >0.9999 | 12 | 100.0 ± 0.0 | vs. YP, >0.9999 | 11 | 100.0 ± 0.0 | vs. YP, 0.0063** |
| YP | 8 | 100.0 ± 0.0 | | 11 | 100.0 ± 0.0 | | 13 | 41.0 ± 10.8 | |
| AU | 7 | 95.2 ± 4.8 | vs. YC, >0.9999 | 19 | 100.0 ± 0.0 | vs. YC, >0.9999 | 15 | 73.34 ± 9.3 | vs. YC, 0.5802 |
| | | | vs. YP, >0.9999 | | | vs. YP, >0.9999 | | | vs. YP, 0.4308 |
| AI | 11 | 72.7 ± 8.8 | vs. AU, 0.2046 | 18 | 85.19 ± 6.2 | vs. AU, 0.0049** | 16 | 12.5 ± 6.0 | vs. AU, 0.0008** |
| | | | vs. YC, 0.0236* | | | vs. YC, 0.0188* | | | vs. YC, <0.0001*** |
| | | | vs. YP, 0.0236* | | | vs. YP, 0.0241* | | | vs. YP, 0.3874 |

| | 20 Hz, 2s, 2 mV below | | | 20 Hz, 250 ms, 2 mV below | | | 20 Hz, 250 ms, 5 mV below | | |
|---|---|---|---|---|---|---|---|---|---|
| Group | Q1 (25%) | Median | Q3 (75%) | Q1 | Median | Q3 | Q1 | Median | Q3 |
| YC | 1.00 | 1.00 | 1.00 | 1.00 | 1.00 | 1.00 | 1.00 | 1.00 | 1.00 |
| YP | 1.00 | 1.00 | 1.00 | 1.00 | 1.00 | 1.00 | 0.00 | 0.33 | 0.83 |
| AU | 1.00 | 1.00 | 1.00 | 1.00 | 1.00 | 1.00 | 0.67 | 1.00 | 1.00 |
| AI | 0.33 | 0.67 | 1.00 | 0.67 | 1.00 | 1.00 | 0.00 | 0.00 | 0.25 |

n, number of cells in group; p-value, Dunn's multiple comparisons test.

**Table 5.** Persistent firing probability evoked with a 100 pA, 150 pA, and 200 pA training stimulus with statistical differences between YC, YP, AU, and AI.
Related to *Figure 2—figure supplement 1*.

| Group | n | 100 pA (%) | p-Value | n | 150 pA (%) | p-Value | n | 200 pA (%) | p-Value |
|---|---|---|---|---|---|---|---|---|---|
| YC | 14 | 100.0 ± 0.0 | vs. YP, >0.9999 | 10 | 100.0 ± 0.0 | vs. YP, >0.9999 | 8 | 100.0 ± 0.0 | vs. YP, >0.9999 |
| YP | 8 | 100.0 ± 0.0 | | 4 | 100.0 ± 0.0 | | 4 | 100.0 ± 0.0 | |
| AU | 17 | 100.0 ± 0.0 | vs. YC, >0.9999 | 11 | 81.8 ± 10.4 | vs. YC, >0.9999 | 7 | 95.2 ± 4.8 | vs. YC, >0.9999 |
| | | | vs. YP, >0.9999 | | | vs. YP, >0.9999 | | | vs. YP, >0.9999 |
| AI | 16 | 68.8 ± 10.3 | vs. AU, 0.0012** | 14 | 57.1 ± 8.9 | vs. AU, 0.1846 | 13 | 61.5 ± 11.8 | vs. AU, 0.2366 |
| | | | vs. YC, 0.0024** | | | vs. YC, 0.0028** | | | vs. YC, 0.0356* |
| | | | vs. YP, 0.0164* | | | vs. YP, 0.0639~ | | | vs. YP, 0.1836 |

| | 100 pA | | | 150 pA | | | 200 pA | | |
|---|---|---|---|---|---|---|---|---|---|
| Group | Q1 (25%) | Median | Q3 (75%) | Q1 | Median | Q3 | Q1 | Median | Q3 |
| YC | 1.00 | 1.00 | 1.00 | 1.00 | 1.00 | 1.00 | 1.00 | 1.00 | 1.00 |
| YP | 1.00 | 1.00 | 1.00 | 1.00 | 1.00 | 1.00 | 1.00 | 1.00 | 1.00 |
| AU | 1.00 | 1.00 | 1.00 | 0.67 | 1.00 | 1.00 | 1.00 | 1.00 | 1.00 |
| AI | 0.33 | 1.00 | 1.00 | 0.33 | 0.50 | 1.00 | 0.17 | 0.67 | 1.00 |

*n*, number of cells in group; *p*-value, Dunn's multiple comparisons test.

The rate of persistent firing was measured using a 250 ms, 20 Hz train of depolarizing current pulses, with the neuronal membrane potential held at 2 mV more hyperpolarized than spontaneous firing threshold (*Figure 4A,C*). To ensure that changes to persistent firing rate and onset latency were a true representation of the effects of aging and learning on these factors independent of firing probability, neurons that were not able to persistently fire (i.e. 0% firing probability) were excluded from these analyses.

Mean firing rate was determined as the average firing rate across the entire sweep. In behaviorally naive animals, the mean rate of persistent firing was significantly reduced in LEC III pyramidal neurons from AN animals compared to those from YN animals (*Figure 4B*; *Table 7*). Firing rate also increased over time (RM ANOVA, $F_{2.244, 67.31}$ = 43.81, p<0.0001), with the firing rate from neurons from YN animals rising faster over time compared to neurons from AN animals ($F_{1,30}$ = 5.687, p=0.0236). Neurons from YN animals reached a peak of 6.45 ± 0.63 spikes/sec at the 9–10 s bin. Neurons from AN animals, however, only reached a peak of 4.20 ± 0.67 spikes/sec at the 19–20 s bin. The young and aged difference in the time to peak firing rate resulted in a significant interaction

**Table 6.** Persistent firing probability with statistical differences in young (YC, YP, YN) and aged (AU, AI, AN).
Related to *Figure 2—figure supplement 2*.

| Group | n | Probability (%) | p-Value |
|---|---|---|---|
| YC | 11 | 100.0 ± 0.0 | |
| YP | 13 | 41.0 ± 10.8 | vs. YC, 0.0062** |
| YN | 21 | 39.7 ± 10.2 | vs. YC, 0.0010** |
| | | | vs. YC, 0.9999 |

| Group | n | Probability (%) | p-Value |
|---|---|---|---|
| AU | 15 | 73.3 ± 9.3 | |
| AI | 16 | 12.5 ± 6.0 | vs. AU, 0.0003*** |
| AN | 14 | 16.7 ± 9.7 | vs. AU, 0.0009*** |
| | | | vs. YC, 0.9999 |

*n*, number of cells in group; *p*-value, Dunn's multiple comparisons test. Persistent firing probability evoked with a 250 ms long 20 Hz training stimulus and membrane holding potential 5 mV more hyperpolarized than spontaneous firing threshold.

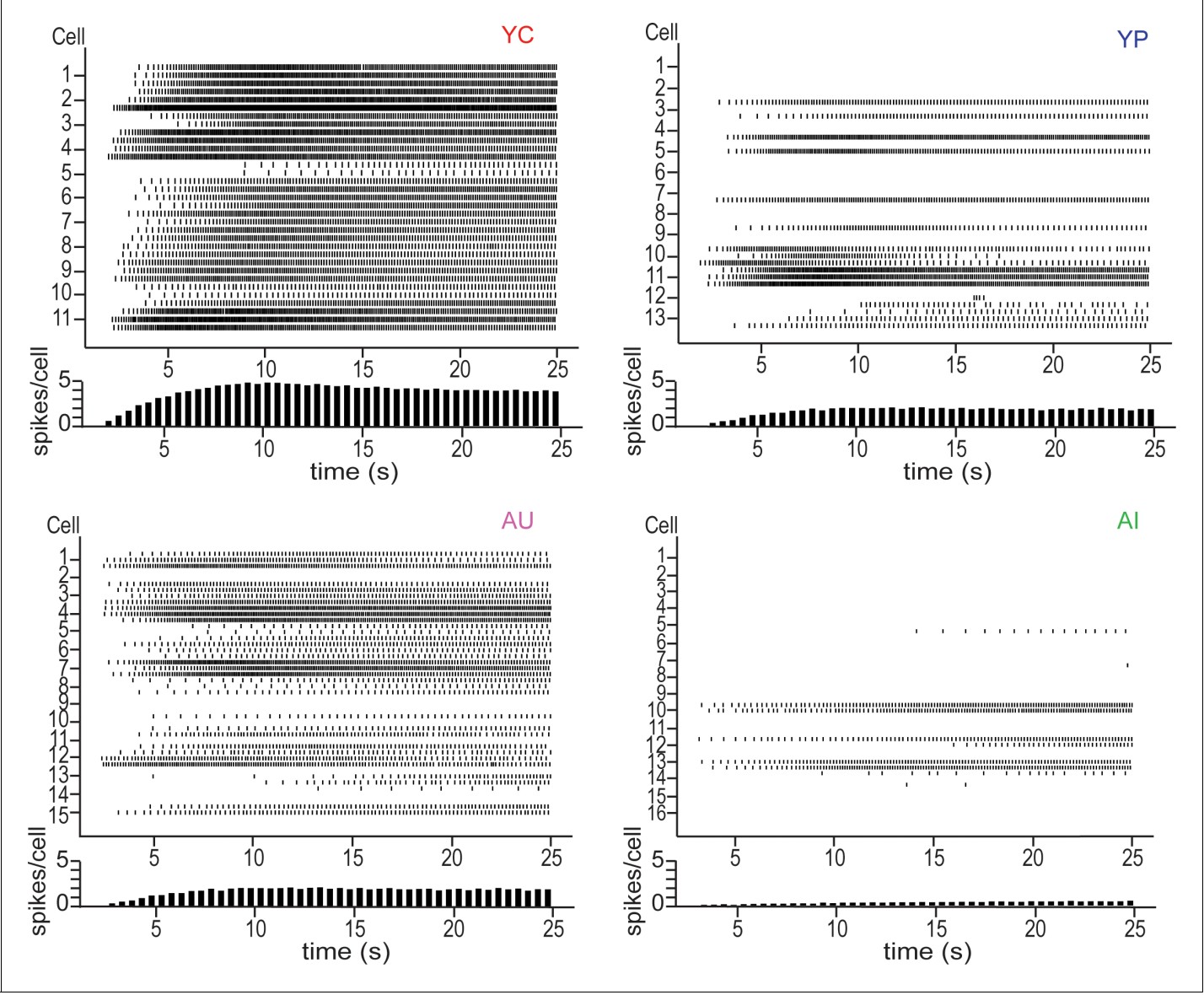

**Figure 3.** Raster plots of persistent firing activity for neurons from YC (*top left*), YP (*top right*), AU (*bottom left*) and AI (*bottom right*) animals. Persistent firing evoked with 250 ms long 20 Hz pulses and neuronal membrane potential held at 5 mV below spontaneous firing. Activity from each of the three sweeps is shown for each neuron. Each vertical line in a row represents a spike. Each row represents one sweep of activity, with each cell marked by its second sweep of activity. Below each raster plot is a histogram of activity across the sweep. Neurons from YC animals are the most active, having the highest spike count per cell across the sweep. In contrast, neurons from AI animals have the lowest spike count across the sweep, reflecting its inactivity.

between time and age group ($F_{22, 660} = 4.063$, $p<0.0001$). A one phase exponential decay function ($Y(t)=Ymax$ $(1 - e^{-t/\tau})$) fit to the rising phase of the firing rate (0–9 s) averaged across all neurons revealed a time constant of 2.66 s for neurons from YN animals, but a time constant of 4.72 s for neurons from AN animals (*Figure 4B*).

Learning also changed the mean rate of persistent firing (ANOVA, $F_{3, 55} = 14.00$, $p<0.0001$). Neurons from YP animals were similar to YN animals (*Figure 4—figure supplement 1*; *Table 8*). However, neurons from YC had a faster rate of firing compared to YP, AU, and AI (*Figure 4D*; *Table 9*). Although the probability of firing in neurons from AU animals had been similar to those of YC animals, neurons from AU animals were not able to fire at the same rate as YC. In fact, the persistent firing rate of neurons from AU animals was not statistically different from neurons of AI animals,

although they did fire faster than neurons from AN animals (*Figure 4—figure supplement 1*; *Table 8*). Therefore, although both groups successfully acquired the trace eyeblink paradigm, persistent firing ability in neurons from AU was not as robust as in YC.

Despite the slower firing rate from the aged group of animals, there was a significant positive correlation between behavior and persistent firing rate amongst the aged group (AU, AI) that was not present amongst the YC group (*Figure 4—figure supplement 2*).

As in the case from behaviorally naive animals, the rate of firing increased over time (RM ANOVA, $F_{2.216, 121.9} = 139.8$, p<0.0001). There was a significant interaction between time and learning group, ($F_{66, 1210} = 2.967$, p<0.0001) indicating that there were also differences in how quickly the rate of firing increased between the groups. Neurons from YC animals reached a peak of $10.92 \pm 0.74$ spikes/sec at the 5–6 s bin. Neurons from YP animals were only able to reach a peak firing rate of $7.76 \pm 0.73$ spikes/sec at the 8–9 s bin, however, indicating that learning caused persistent firing rate to increase faster (*Table 9*). Neurons from AU animals reached a peak firing rate of $7.00 \pm 0.52$ spikes/sec at the 8–9 s bin. The activity of AU neurons was not able to reach the same rate of firing as YC. In fact, the rate of firing across time for AU was significantly slower even compared to the rate of firing from YP (*Table 9*). Despite this, neurons from AU rats were able to fire more quickly than AI, who only reached a peak firing rate of $5.49 \pm 0.54$ spikes/sec at the 10–11 s bin. A one phase exponential decay function fit to the rising phase (0–9 s bins) revealed a time constant of 1.60 s for YC, being the fastest to rise in firing rate and a time constant of 3.46 s for AI, being the slowest to rise in firing rate. The time constants of AU and YP were similar, at 2.50 s and 2.24 s respectively (*Figure 4C,D*). Analysis of the time to the first action potential of persistent firing from the end of the current injection revealed an effect of learning ($F_{3, 55} = 4.260$, p=0.0089) and aging on onset latency. In behaviorally naïve animals, neurons from AN have a slower latency than neurons from YN (*Table 7*). We see this reflected also in AI animals, which have the slowest onset latency. Successful learning, however, decreased latency, resulting in significantly different onset times between YC and AI (*Figure 4E*; *Table 9*).

## The postburst AHP is altered with aging and learning in LEC III pyramidal neurons

So far, we have determined that learning and aging have opposing effects on persistent firing properties of LEC III neurons. Next, we wanted to determine the underlying mechanisms of these changes.

In CA1 neurons, aging has been shown to decrease intrinsic excitability, while successful learning has the opposite effect on neurons from young adult and aged animals (*Moyer et al., 1996*; *Moyer et al., 2000*; *Disterhoft and Oh, 2006*). Given the interplay between intrinsic excitability and aging and learning in the CA1 region, we wanted to determine whether there were similar changes in intrinsic excitability in LEC III, which may contribute to the changes in persistent firing we have observed.

One measure of intrinsic excitability is the postburst afterhyperpolarization (AHP). The postburst AHP is a hyperpolarizing current that mediates a refractory period after a burst of action potentials. It can be divided into the medium (mAHP) and slow (sAHP) components. These two components are mediated by separate $Ca^{2+}$-dependent $K^+$ currents (*Oh et al., 2010*). The mAHP lasts for 50–200 ms after the burst of action potentials and is typically measured at the peak of the AHP. The mAHP is mediated by the apamin-sensitive SK2 channel (*Bond et al., 2004*; *Hammond et al., 2006*; *Disterhoft and Oh, 2006*; *McKay et al., 2012*). The sAHP lasts for several seconds after the burst of action potentials and is measured 1 s after the burst of action potentials. The channels that mediate the sAHP have yet to be identified. However, it is known that it is not mediated by the same apamin-sensitive SK2 channel that mediates the mAHP, as increasing or knocking out SK2 activity has no effect on the sAHP (*Bond et al., 2004*; *Hammond et al., 2006*; *Disterhoft and Oh, 2006*; *McKay et al., 2012*).

In the CA1 region of the hippocampus, the mAHP and sAHP are larger in pyramidal neurons from aged animals compared to young adult animals, indicating decreased excitability (*Power et al., 2002*; *Matthews et al., 2009*). We evoked the postburst AHP in LEC III pyramidal neurons from behaviorally naïve young adult and aged rats by applying a train of 15 suprathreshold current injections (*Figure 5A*). Neurons from AN rats had a larger postburst AHP compared to neurons from YN rats (*Figure 5B*; *Table 10*). Therefore, pyramidal neurons in LEC III undergo an aging-related

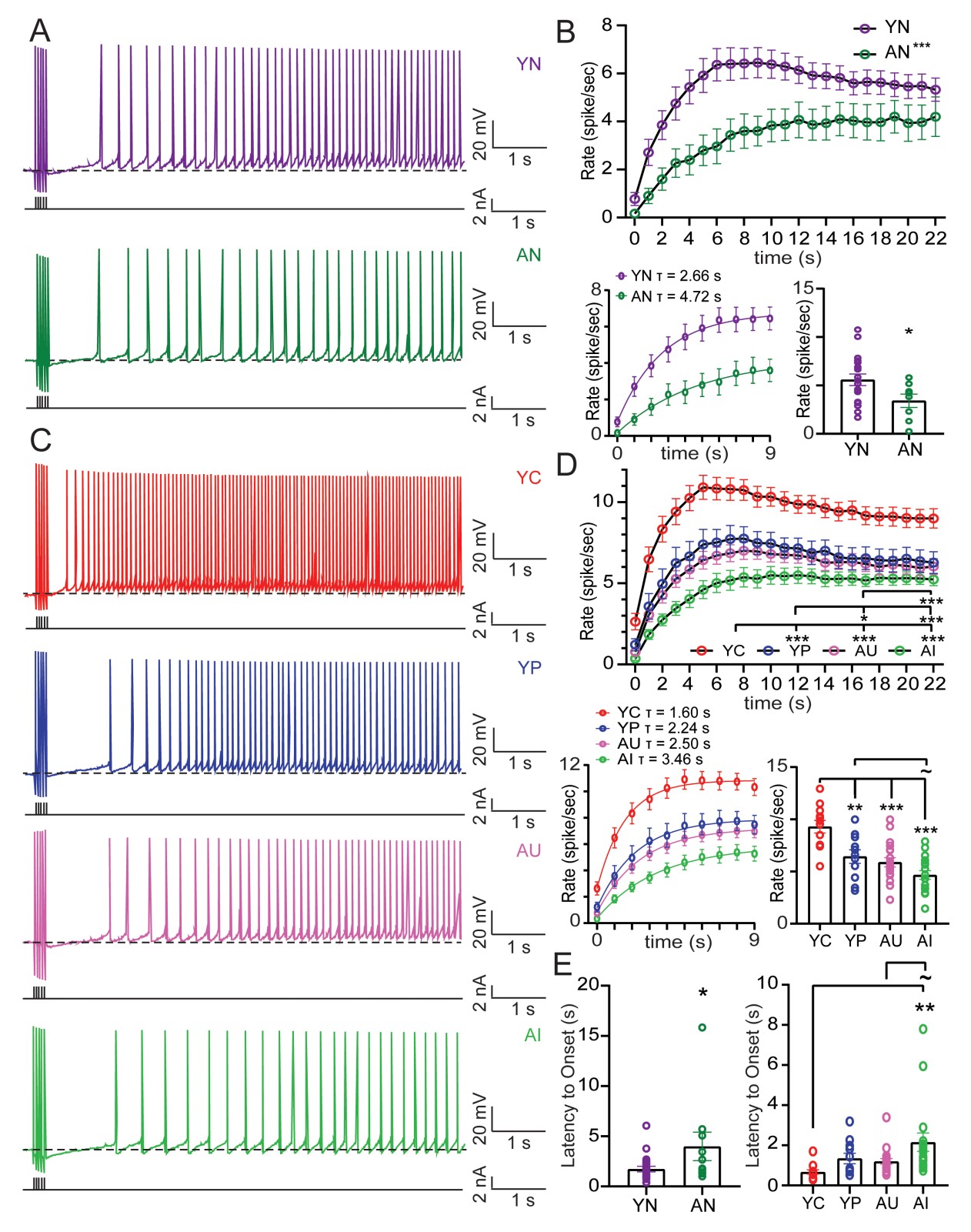

**Figure 4.** Aging decreases persistent firing rate and increases onset latency, while successful learning in young and aged increases firing rate and decreases latency. (**A**) Persistent firing example traces from neurons from YN (*top*) and AN (*bottom*) animals evoked with 250 ms long train of 20 Hz current pulses and neuronal membrane potential held at 2 mV more hyperpolarized than spontaneous firing threshold. Dotted lines indicate spontaneous firing threshold. Black lines underneath persistent firing activity indicate the training stimulus. (**B**) *Top*, Firing rate increases over time, with

*Figure 4 continued on next page*

*Figure 4 continued*

neurons from YN animals (n = 22 neurons) increasing firing rate faster than neurons from AN animals (n = 10). (RM ANOVA, ***p<0.0001). Error bars: mean ± SEM. *Bottom Left*, Rising phase of persistent firing fit with a one phase exponential decay function. Error bars: mean ± SEM. *Bottom Right*, Mean firing rate of neurons (firing rate averaged across the entire sweep) from AN animals is slower than neurons from YN animals. (unpaired t-test, *p=0.0281). Error bars: mean ± SEM. (C) Persistent firing example traces from neurons from YC, YP, AU, AI (*top to bottom*) animals evoked with 250 ms long train of 20 Hz current Pulses. Neuronal membrane potential held at 2 mV below spontaneous firing threshold. Dotted lines indicate spontaneous firing threshold. Black lines underneath persistent firing activity indicate the training stimulus. (D) *Top*, Neurons from YC (n = 12 neurons) animals increase firing rate the fastest. Neurons from AU (n = 19) animals had a slower firing rate compared to neurons from YC and YP (n = 11) animals. Neurons from AI animals (n = 17) fire the slowest. (Tukey's, ***p<0.0001; *p=0.0469). Error bars: mean ± SEM. *Bottom Left*, Rising phase of persistent firing fit with a one phase exponential decay function. Error bars: mean ± SEM. *Bottom Right*, Neurons from YC animals have the fastest mean firing rate. (Tukey's, ***p<0.0001; **p=0.0042;~p = 0.0964). Error bars: mean ± SEM. (E) *Left*, Neurons from AN animals have a longer onset latency than neurons from YN. (un-paired t-test, *p=0.0347). Error bars: mean ± SEM. *Right*, Learning impairments in AI animals increased the time to onset, compared to young animals who successfully learn. (Tukey's, **p=0.0063;~p = 0.0711). Error bars: mean ± SEM. See *Tables 7–9*. Source data files for the firing rate is available in *Figure 4—source data 1*.

The online version of this article includes the following source data and figure supplement(s) for figure 4:

**Source data 1.** Source data for the mean firing rate and firing rate over time.

**Figure supplement 1.** Neurons from YC had increased persistent firing rate compared to YP and YN, and neurons from AU had increased persistent firing rate compared to AN.

**Figure supplement 2.** Session 5 %CR from AU and AI groups is positively correlated with persistent firing rate, but there is no correlation between Session 5 %CR from YC group and persistent firing rate.

reduction in intrinsic excitability, similar to what has been observed in CA1 pyramidal neurons (*Tombaugh et al., 2005*; *Disterhoft and Oh, 2006*; *Power et al., 2002*; *Matthews et al., 2009*).

The intrinsic excitability in LEC III neurons is also altered depending on learning condition (mAHP, one-way ANOVA, $F_{3, 39}$ = 18.55, p<0.0001; sAHP, one-way ANOVA, $F_{3, 39}$ = 8.680, p=0.0002). AI had a larger postburst AHP amplitude compared to AU, YC, and YP. YC had a smaller postburst AHP compared to YP (*Figure 5C,D*; *Table 11*). The postburst AHP amplitude of YP animals were similar to that of behaviorally naïve young adult animals (*Figure 5—figure supplement 1*; *Table 12*). The postburst AHP from AU animals were comparable to both YC and YP (*Figure 5C,D*; *Table 11*). These data indicate that neurons from AI animals are significantly less excitable relative to neurons from AU, YC, and even YP animals. We have also observed a positive correlation between learning ability and postburst AHP amplitude amongst the AU and AI group, indicating the relationship between learning and intrinsic excitability (*Figure 5—figure supplement 2*). The correlation was not present, however, amongst the YC animals.

## Persistent firing properties are dependent on ADP size

Muscarinic activation in LEC III neurons, as is the case in these series of experiments, has been demonstrated to eliminate the postburst AHP. Following a burst of action potentials, then, instead of an AHP, there is a depolarization of the membrane potential above the baseline. This is known as the afterdepolarization (ADP) (*Figure 6A,C*). Following the peak of the ADP, the depolarized membrane potential is usually sustained above the baseline membrane potential. This sustained depolarization is known as the plateau potential (PP) (*Fraser and MacVicar, 1996*; *Wu et al., 2004*; *Tahvildari et al., 2007*). The ADP allows the neuron to reach spontaneous firing threshold during an up-state and fire action potentials, while the PP allows the firing to persist. We examined the ADP and PP in LEC III neurons using a 250 ms train of 20 Hz (2 ms, 2 nA) depolarizing current pulses and held the neuronal membrane potential at 10 mV more hyperpolarized than spontaneous firing

**Table 7.** Mean persistent firing rate, peak firing rate, and onset latency with statistical differences between YN and AN. Related to *Figure 4B, E*.

| Group | n | Mean Firing Rate (spikes/sec) | p-Value | Peak Firing Rate (spikes/sec) | Latency to Onset (s) | p-Value |
|---|---|---|---|---|---|---|
| YN | 22 | 5.36 ± 0.54 | | 6.45 ± 0.63 | 1.74 ± 0.27 | |
| AN | 10 | 3.29 ± 0.58 | vs. YN, 0.0281* | 4.20 ± 0.67 | 3.96 ± 1.41 | vs. YN, 0.0347* |

*n*, number of cells in group; *p*-value, unpaired t-test.

**Table 8.** Mean persistent firing rate, peak firing rate, and onset latency with statistical differences between YC, YP, AU, AI. Related to *Figure 4D, E*.

| Group | n | Mean Firing Rate (spikes/sec) | p-Value | Peak Firing Rate (spikes/sec) | Firing Rate Across Time *p*-value | Latency to Onset (s) | p-Value |
|---|---|---|---|---|---|---|---|
| YC | 12 | 9.30 ± 0.60 | vs. YP, 0.0042** | 10.92 ± 0.74 | vs. YP, <0.0001*** | 0.67 ± 0.07 | vs. YP, 0.5026 |
| YP | 11 | 6.43 ± 0.65 | | 7.76 ± 0.73 | | 1.35 ± 0.20 | |
| AU | 19 | 5.88 ± 0.44 | vs. YC, <0.0001*** | 7.00 ± 0.52 | vs. YC, <0.0001*** | 1.20 ± 0.11 | vs. YC, 0.6059 |
| | | | vs. YP, 0.8783 | | vs. YP, 0.0469* | | vs. YP, 0.9861 |
| AI | 17 | 4.66 ± 0.42 | vs. AU, 0.2427 | 5.49 ± 0.54 | vs. AU, <0.0001*** | 2.15 ± 0.46 | vs. AU, 0.0711~ |
| | | | vs. YC, <0.0001*** | | vs. YC, <0.0001*** | | vs. YC, 0.0063** |
| | | | vs. YP, 0.0964~ | | vs. YP, <0.0001*** | | vs. YP, 0.2743 |

n, number of cells in group; p-value, Tukey's multiple comparisons test.

threshold, to preclude persistent firing. The peak amplitude of the ADP was used to determine the size of the ADP. We also measured the size of the ADP and PP by measuring the area under the curve of the membrane depolarization.

In behaviorally naive animals, the ADP peak amplitude evoked from neurons of AN rats was smaller compared to the peak of the ADP evoked from neurons of YN animals. The area of the ADP and PP was also revealed to be smaller for neurons from aged animals (*Figure 6B*; *Table 13*).

Neurons from YC rats were consistently successful in their persistent firing ability, even when the membrane potential was 5 mV more hyperpolarized than spontaneous firing threshold. Therefore, it is not surprising that when the ADP was evoked from YC neurons (250 ms train of 20 Hz, 10 mV more hyperpolarized than spontaneous firing), the peak amplitude of the ADP from YC neurons was largest. The area of the ADP and PP was also largest in YC. (Amplitude: one-way ANOVA, $F_{3,39}$ = 29.73, p<0.0001) (AUC: one-way ANOVA $F_{3,39}$ = 15.88, p<0.0001) (*Figure 6C,D*; *Table 14*) (*Figure 6—figure supplement 1*; *Table 15*). In contrast, neurons from AI animals, having consistently the lowest probability of persistent firing compared to YC, YP, and AU, have ADPs that are smallest in terms of amplitude. The area of their ADP and PP was also smallest (*Figure 6C,D*; *Table 14*). The ADP size of neurons from AI animals was similar to neurons from AN animals (*Figure 6—figure supplement 1*; *Table 15*).

Further analyses revealed that ADP size had an effect on the probability, onset latency, and persistent firing rate. The ADP amplitude heavily influenced whether or not a neuron was able to persistently fire when the membrane potential was held at 5 mV below spontaneous firing. An ADP

**Table 9.** Persistent firing rate with statistical differences in young (YC, YP, YN) and aged (AU, AI, AN). Related to *Figure 4—figure supplement 1*.

| Group | n | Mean firing rate (spikes/sec) | p-Value |
|---|---|---|---|
| YC | 12 | 9.30 ± 0.60 | |
| YP | 11 | 6.43 ± 0.65 | vs. YC, 0.0144* |
| YN | 22 | 5.36 ± 0.54 | vs. YC, <0.0001*** |
| | | | vs. YP, 0.4406 |
| Group | n | Mean firing rate (spikes/sec) | p-Value |
| AU | 19 | 5.88 ± 0.43 | |
| AI | 17 | 4.66 ± 0.42 | vs. AU, 0.1216 |
| AN | 10 | 3.29 ± 0.58 | vs. AU, 0.0020** |
| | | | vs. AI, 0.1536 |

n, number of cells in group; p-value, Tukey's multiple comparisons test.

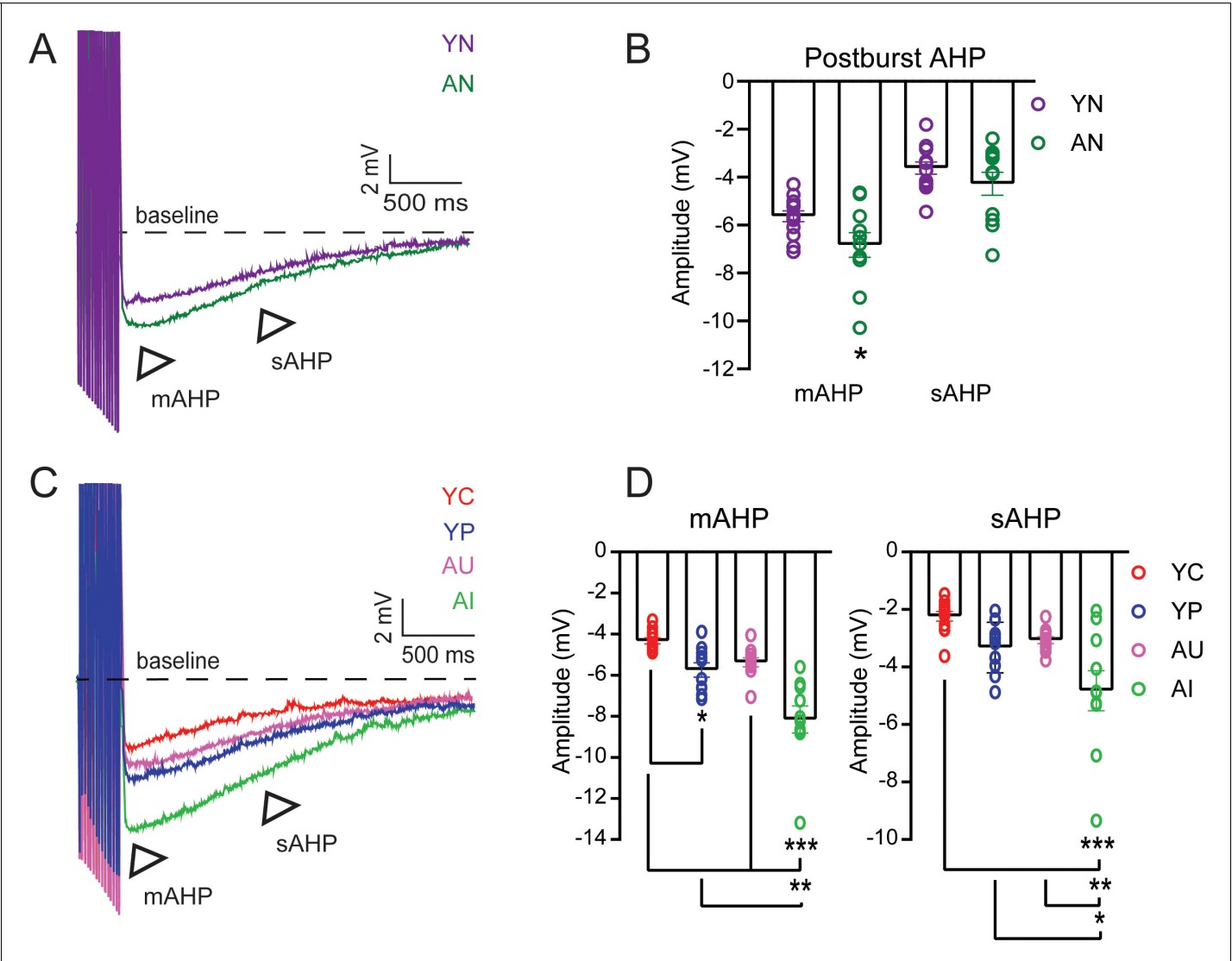

**Figure 5.** The postburst AHP amplitude is increased in LEC layer III neurons from behaviorally naïve aged animals but is decreased in neurons from both young adult and aged animals after successful acquisition of trace eyeblink conditioning. (**A**) Postburst AHP example traces from neurons from YN (purple) and AN (dark green) animals. Arrows indicate the medium (mAHP) and slow (sAHP) AHPs. Suprathreshold current injections have been truncated to illustrate the postburst AHP. (**B**) mAHP from cells from AN animals (n = 11 neurons) is larger in amplitude relative to cells from YN animals (n = 13). (un-paired t-test, *p=0.0343). No difference between neurons from aged and young animals in sAHP. Error bars: mean ± SEM. (**C**) Postburst AHP Example Traces from Neurons from YC, YP, AU, AI. Arrows indicate the medium (mAHP) and slow (sAHP) AHPs. (**D**) Successful learning decreases Postburst AHP amplitude. Error bars: mean ± SEM. *Left*, mAHP Amplitudes. Neurons from YC (n = 12 neurons, n = 9 rats) animals have a smaller mAHP relative to neurons from YP (n = 10, n = 8 rats) animals. Neurons from AU animals (n = 11, n = 8 rats) are comparable to the neurons from young animals, while neurons from AI animals (n = 10, n = 5 rats) have a larger mAHP amplitude relative to neurons from YC, YP, and AU animals. (Tukey's, ***p<0.0001; **p=0.0004; *p=0.0493). *Right*, sAHP Amplitudes. AI have a larger sAHP amplitude relative to the others. (Tukey's, ***p<0.0001; **p=0.0092; *p=0.0380. Error bars: mean ± SEM. See *Tables 10–12*. Source data files for the postburst AHP amplitudes is available in *Figure 5— source data 1*.

The online version of this article includes the following source data and figure supplement(s) for figure 5:

**Source data 1.** Source data for the postburst AHP.

**Figure supplement 1.** Neurons from YC had smaller mAHP and sAHP amplitude compared to YP and YN, and neurons from AU had smaller mAHP and sAHP amplitude compared to AI.

**Figure supplement 2.** Session 5 %CR from AU and AI groups is positively correlated with mAHP and sAHP amplitudes, but there is no correlation between Session 5 %CR from YC group and mAHP and sAHP amplitudes.

**Table 10.** Postburst AHP (mAHP and sAHP) values with statistical differences between YN and AN. Related to *Figure 5B*.

| Group | n | mAHP (mV) | p-Value | sAHP (mV) | p-Value |
|-------|---|-----------|---------|-----------|---------|
| YN | 13 | −5.63 ± 0.23 | | −3.62 ± 0.26 | |
| AN | 11 | −6.83 ± 0.51 | vs. YN, 0.0343* | −4.28 ± 0.48 | vs. YN, 0.2167 |

*n*, number of cells in group; p-value, unpaired t-test.

amplitude of ~4 mV was the boundary which determined successful or unsuccessful firing, with amplitudes at or above 4 mV allowing a neuron to successfully persistently fire (*Figure 6E*). The ADP amplitude was also positively correlated with mean persistent firing rate (Pearson, r = 0.6265, p<0.0001), indicating that a larger ADP size allowed for the neuron to fire more quickly. Finally, ADP amplitude was also negatively correlated with onset latency (Pearson, r = −0.4291, p=0.0029), indicating that a larger ADP size allowed the neuron to start firing earlier in the sweep (*Figure 6E*).

Just as we observed with persistent firing rate and postburst AHP amplitude, the ADP size is significantly positively correlated with learning ability amongst AU and AI animals, but not amongst YC animals (*Figure 6—figure supplement 2*). Amongst the aged group, then, learning ability may help determine intrinsic excitability, the size of the ADP, and subsequently, persistent firing. Amongst the YC animals, there may not be a significant correlation because the young animals trained on trace eyeblink conditioning successfully learned the task.

## The ADP and PP in behaviorally naive young adult and YC animals are Ca²⁺ dependent

Previous studies have shown that the ADP and PP that generate persistent firing in LEC III and V are mediated by an activity-dependent $Ca^{2+}$ influx into the neuron (*Tahvildari et al., 2008*). The initial current injection to evoke persistent firing generates an influx of $Ca^{2+}$ into the cell body, which may activate cation currents, including a regenerative current that underlies the PP (*Fraser and MacVicar, 1996*). To determine whether the enhanced ADP in learning is also dependent on $Ca^{2+}$, we bath applied the voltage-gated $Ca^{2+}$ channel blocker, $CdCl_2$, to neurons from YN and YC animals. We evoked the ADP and PP by holding the membrane potential 10 mV more hyperpolarized than spontaneous firing and found that the ADP and PP were eliminated in neurons from YN animals as well as in YC (ADP: RM ANOVA, pre-post $F_{1,6}$ = 28.09, p=0.0018; pre-post x group $F_{1,6}$ = 4.521, p=0.0776) (AUC: pre-post RM ANOVA, $F_{1,6}$ = 64.31, p=0.0002; pre-post x group $F_{1,6}$ = 18.64, p=0.0050) (*Figure 7*; *Table 16*). Abolishment of the ADP and PP eliminated persistent firing, as evoked by a 250 ms long 20 Hz train of current pulses (Wilcoxon, W = −36.00, p=0.0078). These data suggest that, like neurons of YN animals, the enhanced ADP and PP of LEC III neurons from YC are also dependent on $Ca^{2+}$.

**Table 11.** Postburst AHP (mAHP and sAHP) values with statistical differences between YC, YP, AU, and AI. Related to *Figure 5D*.

| Group | n | mAHP (mV) | p-Value | sAHP (mV) | p-Value |
|-------|---|-----------|---------|-----------|---------|
| YC | 12 | −4.33 ± 0.15 | vs. YP, 0.0493* | −2.24 ± 0.17 | vs. YP, 0.1653 |
| YP | 10 | −5.74 ± 0.35 | | −3.32 ± 0.28 | |
| AU | 11 | −5.38 ± 0.22 | vs. YC, 0.1912 | −3.07 ± 0.12 | vs. YC, 0.3590 |
| | | | vs. YP, 0.9016 | | vs. YP, 0.9606 |
| AI | 10 | −8.16 ± 0.66 | vs. AU, <0.0001*** | −4.82 ± 0.70 | vs. AU, 0.0092** |
| | | | vs. YC, <0.0001*** | | vs. YC, <0.0001*** |
| | | | vs. YP, 0.0004** | | vs. YP, 0.0380* |

*n*, number of cells in group; p-value, Tukey's multiple comparisons test.

**Table 12.** Postburst AHP amplitude with statistical differences in young (YC, YP, YN) and aged (AU, AI, AN). Related to *Figure 5—figure supplement 1*.

| Group | n | mAHP (mV) | p-Value | sAHP (mV) | p-Value |
|---|---|---|---|---|---|
| YC | 12 | −4.33 ± 0.15 | | −2.24 ± 0.17 | |
| YP | 10 | −5.74 ± 0.35 | vs. YC, 0.0011** | −3.32 ± 0.28 | vs. YC, 0.0098** |
| YN | 13 | −5.63 ± 0.23 | vs. YC, 0.0013** | −3.62 ± 0.26 | vs. YC, 0.0005*** |
| | | | vs. YP, 0.9443 | | vs. YP, 0.6658 |
| Group | n | mAHP (mV) | p-Value | sAHP (mV) | p-Value |
| AU | 11 | -5.38 ± 0.22 | | -3.07 ± 0.12 | |
| AI | 10 | -8.16 ± 0.66 | vs. AU, 0.0011** | -4.82 ± 0.70 | vs. AU, 0.0394* |
| AN | 11 | -6.83 ± 0.51 | vs. AU, 0.0985~ | -4.28 ± 0.48 | vs. AU, 0.1801 |
| | | | vs. AI, 0.1538 | | vs. AI, 0.7068 |

n, number of cells in group; p-value, Tukey's multiple comparisons test.

## Discussion

In this study, we demonstrated that learning and aging have opposing effects on persistent firing in LEC III. In behaviorally naive animals, persistent firing probability was decreased in LEC III pyramidal neurons from aged animals. On the other hand, under conditions which did not evoke robust firing in neurons from behaviorally naïve or control animals, successful learning enhanced firing probability in neurons from both young adult and aged animals. The rate of persistent firing was also affected, with aging and learning again having opposing effects. We determined that there were also learning and aging-related changes to the postburst AHP, a measure of intrinsic excitability. The changes to the postburst AHP may contribute to the changes to persistent firing that we have observed. We also observed aging and learning-related changes to the afterdepolarization (ADP). The ADP size was highly correlated with persistent firing rate and onset latency. In neurons from both YN and YC animals, the ADP was eliminated with a $Ca^{2+}$ channel blocker, implying that not only is the ADP activated by a $Ca^{2+}$ dependent mechanism, as others have reported, but also that successful learning enhances this mechanism to allow for the enhanced persistent firing ability we have observed.

Although the persistent firing of neurons from aged animals was less robust, a learning-related increase in the size of the ADP in neurons from both aged and young adult animals allowed for the gap between a more hyperpolarized membrane holding potential and spontaneous firing threshold to be more easily overcome than in control animals. Functionally, this may allow for persistent firing to occur more easily following learning, which may be important for the expression and consolidation of the memory.

The other effect of a learning-related increase in ADP size is to increase the rate of persistent firing within the theta range. Theta-modulation in the LEC has been shown to be relevant for task-specific actions, such as running (*Deshmukh et al., 2010*). A previous study has also shown theta-locking of LEC III neurons when the animal attends to sensory cues during an associative learning task (*Igarashi et al., 2014*). It has been hypothesized that persistent firing rate could provide a phase code for changes to object features in the continuous dimension, such that an increase in persistent firing rate could allow for a phase shift in firing to encode continuous dimensions (*Hasselmo et al., 2010*). In this manner, an increase in persistent firing rate during trace eyeblink conditioning could allow for the properties of the stimuli (e.g. tone, timing) to be properly encoded so that a more precise behavioral response can be elicited.

Previous studies have also indicated the functional importance of LEC theta-coupling with the hippocampus and other memory-associated regions, such as the prefrontal cortex (*Takehara-Nishiuchi et al., 2011*). The LEC has been shown to increase theta synchronization with both the hippocampus and the prefrontal cortex following CS onset during the acquisition phase of trace eyeblink conditioning. Theta oscillations between the LEC and the prefrontal cortex, in particular, become strongly synchronized as the animal acquires trace eyeblink conditioning, and continue this synchronized activity through encoding and consolidation (*Takehara-Nishiuchi et al., 2011*). In fact, lesions to the temporoammonic pathway have been shown to eliminate consolidation of a

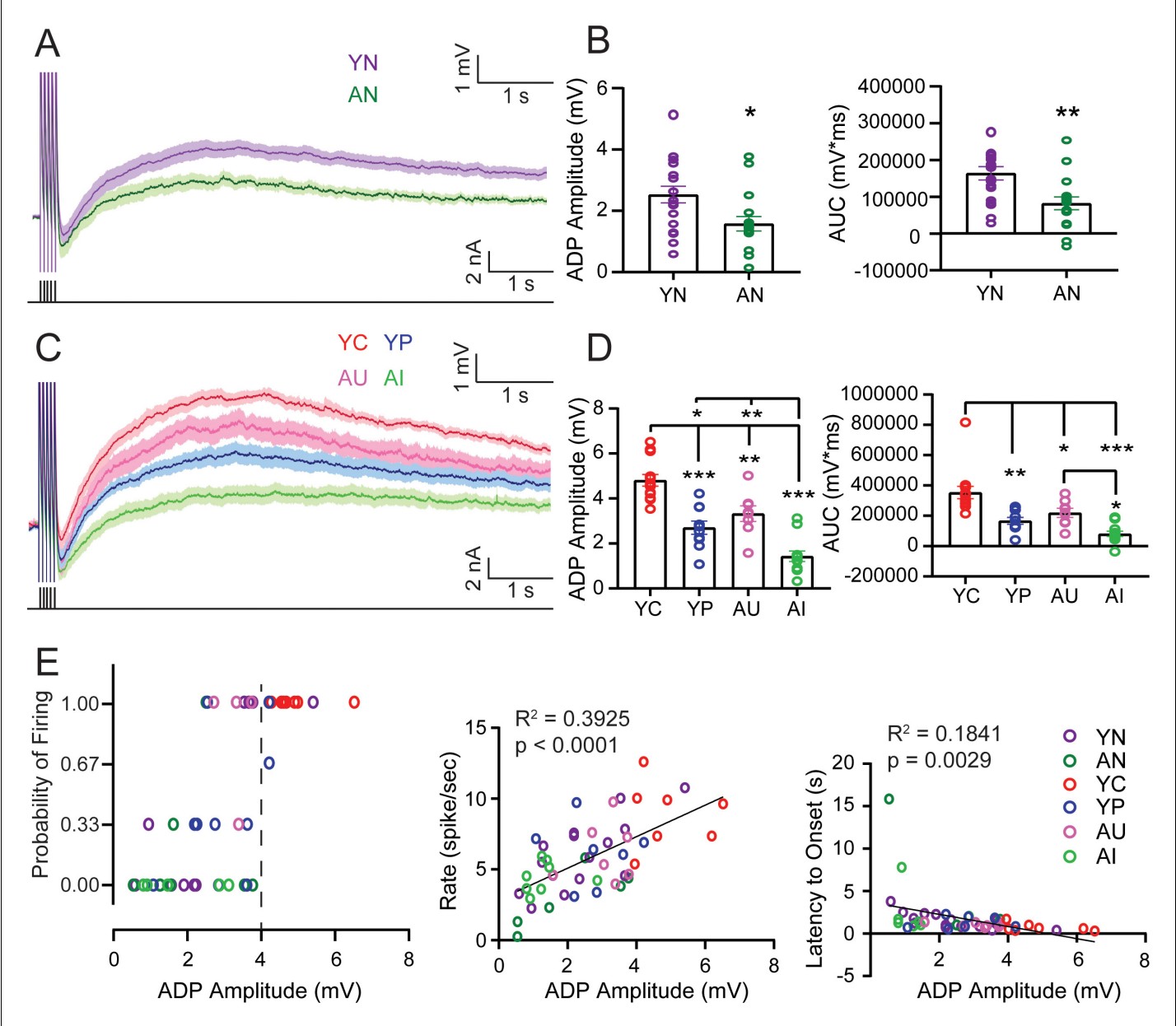

**Figure 6.** The afterdepolarization (ADP) and plateau potential (PP) determine persistent firing properties and are increased in neurons from young adult and aged animals that successfully acquire trace eyeblink conditioning but decreased in neurons from aged animals that are learning impaired. (**A**) Averaged ADPs from neurons from YN and AN animals. Shaded areas: SEM. Black lines underneath the averaged traces indicate the training stimulus. (**B**) *Left*, ADP peak amplitude from neurons from AN animals (n = 17 neurons) is smaller than in neurons from YN animals (n = 19). (unpaired t-test, *p=0.0130). Error bars: mean ± SEM. *Right*, Neurons from AN animals have a smaller ADP and PP area. (unpaired t-test, **p=0.0025). Error bars: mean ± SEM. (**C**) Averaged ADPs from neurons from YC, YP, AU, and AI animals. Shaded areas: SEM. Black lines underneath the averaged traces indicate the training stimulus. (**D**) *Left*, Neurons from YC animals (n = 13 neurons) have the largest ADP peak amplitude compared to neurons from YP (n = 10), AU (n = 8), and AI (n = 12), while neurons from AI have the smallest ADP peak. (Tukey's, ***p<0.0001; ** YC vs. AU p=0.0042, ** AI vs. AU p=0.0003; *p=0.0115). Error bars: mean ± SEM. *Right*, Neurons from YC animals have the largest ADP and PP area, while neurons from AI animals have the smallest. (Tukey's, ***p<0.0001; **p=0.0005; YC vs. AU *p=0.0265; AI vs. AU *p=0.0231). Error bars: mean ± SEM. (**E**) ADP Amplitude determines persistent firing properties. *Left*, A larger ADP (>~4 mV) allows for more successful persistent firing, when persistent firing was evoked with the membrane potential 5 mV more hyperpolarized than spontaneous firing threshold. Dotted line indicates the ADP size boundary that results in the difference between successful firing (i.e. 1.0) and unsuccessful firing (i.e. 0.0). *Middle*, ADP amplitude is positively correlated with persistent firing rate. *Right*, ADP amplitude is negatively correlated with the latency to firing onset. See *Tables 13–15*. Source data files for the ADP amplitude and AUC is available in *Figure 6—source data 1*.

The online version of this article includes the following source data and figure supplement(s) for figure 6:

*Figure 6 continued on next page*

*Figure 6 continued*

**Source data 1.** Source data for ADP amplitude and AUC.
**Figure supplement 1.** Neurons from YC had larger ADP and PP compared to YP and YN, and neurons from AU had larger ADP and PP compared to AI and AN.
**Figure supplement 2.** Session 5 %CR from AU and AI groups is positively correlated with ADP amplitude, but there is no correlation between Session 5 %CR from the YC group and ADP amplitude.

hippocampus-dependent memory (**Remondes and Schuman, 2004**). An increase in persistent firing rate, therefore, may also be the result of successful acquisition of trace eyeblink conditioning and may support consolidation of the newly acquired association into long-term memory. Aging, however, had the effect of decreasing persistent firing properties such as probability and rate of firing. We observed that neurons from aged animals that were trained on trace eyeblink conditioning but failed to learn the paradigm had a reduced capacity for persistent firing even compared to neurons from young adult control animals (i.e. YP). An aging-related decrease in persistent firing ability may, therefore, contribute to the learning impairments that we and others have observed in the aging population (**Knuttinen et al., 2001**).

At the same time, although neurons from AU animals were able to persistently fire, they were not able to fire as robustly as neurons from YC animals. When the gap between the neuronal membrane holding potential and spontaneous firing threshold was increased, neurons from aged animals that successfully learned were unable to reach the ceiling effect that neurons from young animals that successfully learned were able to reach. Their firing rate was also slower than those from young animals that learned. They also had a slower time to reach peak firing rate compared to neurons from YC animals. A slower time to reach peak firing may make it more difficult for neurons to be included in an ensemble to help bridge the trace interval. Alternatively, it may also result in desynchronization within the LEC network of theta and other rhythms that are necessary for hippocampus-dependent learning, such as gamma (**Igarashi et al., 2014**).

As a result, although neurons from unimpaired aged animals were still able to persistently fire, they were also affected by the changes in neuronal properties that are associated with aging. This leads to their reduced capacity for persistent firing as compared to their young adult counterparts. The ability of AU animals to achieve the same level of learning behaviorally may be the result of compensatory mechanisms in other regions. These regions may include those that are functionally connected to the LEC, such as the prefrontal cortex and the hippocampus.

In the CA1 region of the hippocampus, for example, if an aged animal is able to acquire a hippocampus-dependent task such as trace eyeblink conditioning, the excitability of the neurons increases (**Moyer et al., 2000**; **Matthews et al., 2009**). Moreover, the excitability of these neurons is comparable to those of a young animal who successfully acquired trace eyeblink conditioning, unlike in neurons of LEC III. The firing ability of CA1 neurons, therefore, may serve to help compensate for the less robust persistent firing in LEC III neurons of AU animals. It also seems that the excitability of CA1 neurons may be a better determinant of successful learning in aging than that of LEC III neurons. The differences in excitability changes across separate regions of an interconnected memory network point to individually weighted contributions to the development of hippocampus-dependent memory. Although there may be excitability differences in each region as a result of aging, they may not be equal contributors to memory acquisition. The changes we have observed in excitability in LEC III are also reflected in the AHP. While learning increased the ADP, it decreased the AHP. Aging, on the other hand, had the effect of decreasing the ADP and increasing the AHP. As a

**Table 13.** ADP amplitude and AUC of the ADP and PP with statistical differences between YN and AN. Related to *Figure 6B*.

| Group | n | ADP Amplitude (mV) | p-Value | AUC (mV*ms) | p-Value |
|---|---|---|---|---|---|
| YN | 19 | 2.53 ± 0.27 | | 163800 ± 18081 | |
| AN | 17 | 1.58 ± 0.23 | vs. YN, 0.0130* | 82080 ± 17108 | vs. YN, 0.0025** |

*n*, number of cells in group; p-value, unpaired t-test.

**Table 14.** ADP amplitude and AUC of the ADP and PP with statistical differences between YC, YP, AU, and AI. Related to *Figure 6D*.

| Group | n | ADP Amplitude (mV) | p-Value | AUC (mV*ms) | p-Value |
|---|---|---|---|---|---|
| YC | 13 | 4.81 ± 0.26 | vs. YP, <0.0001*** | 353696 ± 41070 | vs. YP, 0.0005** |
| YP | 10 | 2.70 ± 0.29 | | 166842 ± 22232 | |
| AU | 8 | 3.32 ± 0.35 | vs. YC, 0.0042** | 219628 ± 30188 | vs. YC, 0.0265* |
| | | | vs. YP, 0.4802 | | vs. YP, 0.6918 |
| AI | 12 | 1.43 ± 0.24 | vs. AU, 0.0003** | 80930 ± 17555 | vs. AU, 0.0231* |
| | | | vs. YC, <0.0001*** | | vs. YC, <0.0001*** |
| | | | vs. YP, 0.0115* | | vs. YP, 0.2119 |

*n*, number of cells in group; p-value, Tukey's multiple comparisons test.

result, an inverse relationship between the AHP and the ADP may exist in LEC III pyramidal neurons, with a smaller AHP (i.e. following learning) allowing for the development of a larger ADP, and a larger AHP (i.e. as a result of aging) impeding the development of a larger ADP. A previous study performed in CA1 neurons has in fact shown that elimination of the postburst AHP allows for the development of an ADP. The presence of the AHP inhibited the temporal summation of the mechanism supporting the ADP (*Wu et al., 2004*).

Although the exact channels and mechanisms that underlie the ADP are still unknown, what is known is that both the AHP and ADP are mediated by $Ca^{2+}$ influx following a burst of action potentials. Postburst $Ca^{2+}$ influx, via voltage-gated $Ca^{2+}$ channels, activates $Ca^{2+}$ activated $K^+$ channels, which generate the AHP (*Bond et al., 2004*; *Disterhoft and Oh, 2006*; *McKay et al., 2012*). However, if the AHP is suppressed, as with neuromodulators such as acetylcholine, $Ca^{2+}$-activated cationic conductances (e.g. CAN channels), also activated by postburst $Ca^{2+}$ influx, may present themselves, resulting in the development of an ADP (*Tahvildari et al., 2008*). Further experiments are needed to elucidate the interaction between the AHP and the ADP in LEC III pyramidal neurons, however, and how learning and aging affect both mechanisms in order to support and inhibit persistent firing, respectively.

Finally, although we have established that learning and aging differentially affect persistent firing, the functional role that persistent firing plays during trace eyeblink conditioning and other temporal associative tasks is yet to be determined in vivo. The answer to this question is beyond the scope of the current study, although there are a few possibilities that may be considered.

One possible role is to enhance the activity of their direct downstream target, CA1 pyramidal neurons. Temporoammonic activity from LEC III to CA1 has the ability to modulate Schaffer-Collateral (SC) evoked CA1 spiking activity (*Remondes and Schuman, 2002*). As such, persistent firing in

**Table 15.** ADP Amplitude and AUC of the ADP and PP with statistical differences in young (YC, YP, YN) and aged (AU, AI, AN). Related to *Figure 6—figure supplement 1*.

| Group | n | ADP Amplitude (mV) | p-Value | AUC (mV*ms) | p-Value |
|---|---|---|---|---|---|
| YC | 13 | 4.81 ± 0.26 | | 353696 ± 41070 | |
| YP | 10 | 2.70 ± 0.29 | vs. YC, <0.0001*** | 166842 ± 22232 | vs. YC, 0.0003** |
| YN | 19 | 2.53 ± 0.27 | vs. YC, <0.0001*** | 163800 ± 18081 | vs. YC, <0.0001*** |
| | | | vs. YP, 0.9069 | | vs. YP, 0.9969 |
| Group | n | ADP Amplitude (mV) | p-Value | AUC (mV*ms) | p-Value |
| AU | 8 | 3.32 ± 0.35 | | 219628 ± 30188 | |
| AI | 12 | 1.43 ± 0.23 | vs. AU, 0.0002** | 80930 ± 17555 | vs. AU, 0.0004** |
| AN | 17 | 1.58 ± 0.23 | vs. AU, 0.0003** | 82080 ± 17108 | vs. AU, 0.0002** |
| | | | vs. AI, 0.9063 | | vs. AI, 0.9990 |

*n*, number of cells in group; p-value, Tukey's multiple comparisons test.

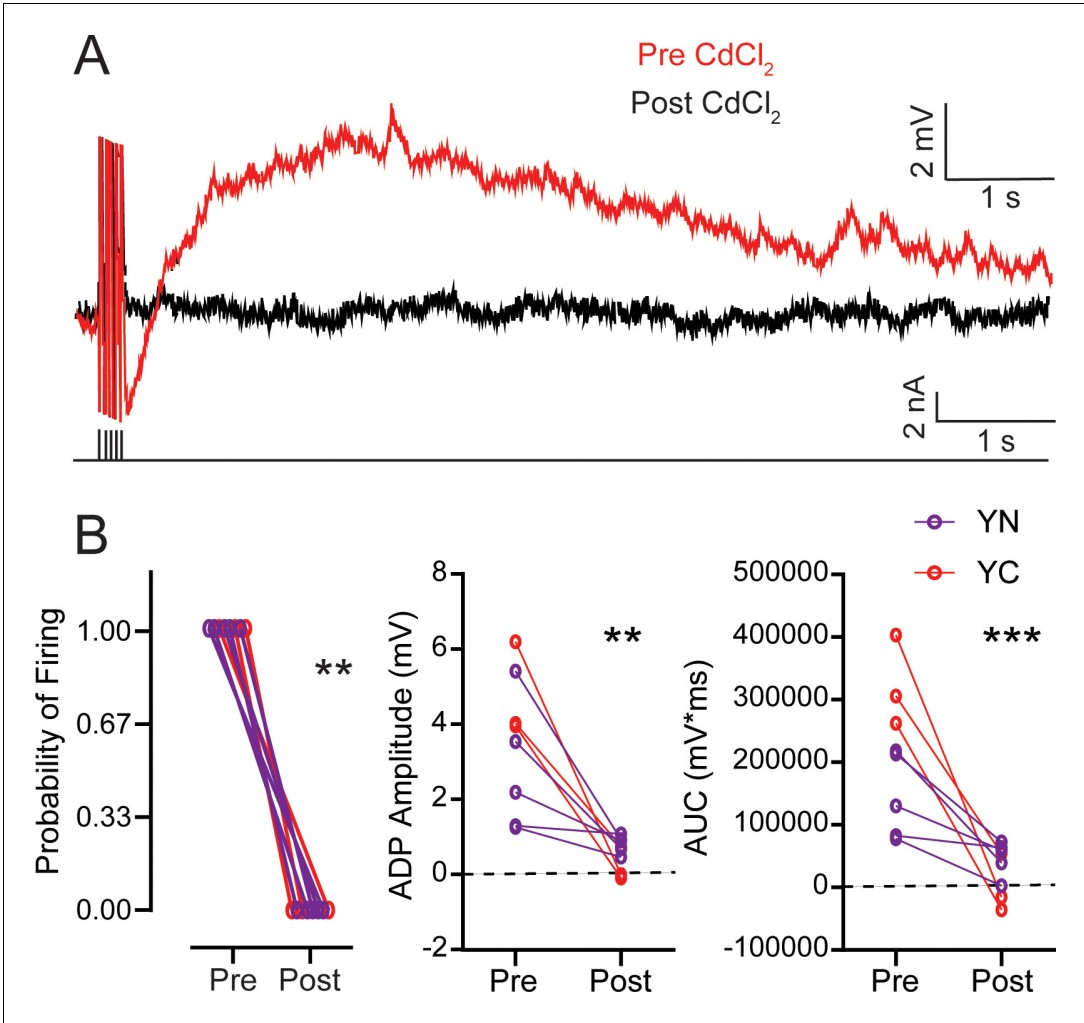

**Figure 7.** The ADP is $Ca^{2+}$-dependent and is eliminated by $CdCl_2$ in neurons from both YN and YC animals. (A) Pre- and post-$CdCl_2$ ADP example traces. ADP is evoked with 250 ms 20 Hz train of current pulses as the membrane potential is held 10 mV more hyperpolarized than spontaneous firing threshold. Black lines underneath the traces the training stimulus. (B) *Left*, $CdCl_2$ eliminates persistent firing from neurons from both YN (n = 5 neurons) and YC (n = 3) animals. (Wilcoxon, \*\*p=0.0078). *Middle*, $CdCl_2$ reduces the amplitude of the ADP. (pre-post RM ANOVA, \*\*p=0.0018). *Right*, $CdCl_2$ reduces the area under the curve of the ADP and PP. (pre-post RM ANOVA, \*\*\*p=0.0002). See *Table 16*. Source data files for the ADP amplitude and AUC is available in *Figure 7—source data 1*.

The online version of this article includes the following source data for figure 7:

**Source data 1.** Source data for ADP amplitude and AUC before and after $CdCl_2$.

LEC III neurons may enhance SC input to CA1 neurons, allowing for CA1 neurons to be more responsive to convergent CS and US information that is provided via the tri-synaptic pathway. Evidence for this comes from previous studies from our laboratory, which have shown an increase in CA1 activity in response to both the CS and the US in vivo (*McEchron and Disterhoft, 1997*; *McEchron and Disterhoft, 1999*). Therefore, persistent firing in LEC III may serve to enhance the processing of salient stimuli within CA1 neurons, allowing for the successful acquisition of trace eye-blink conditioning.

The prevailing hypothesis in the literature, however, is that persistent firing supports a 'memory' of the CS during the trace interval, so that the association between the CS and the US can be made (*Kitamura et al., 2015*; *Zylberberg and Strowbridge, 2017*). Previous in vivo work has shown that persistent activity in the prefrontal cortex bridges the delay period during a working memory task,

**Table 16.** ADP Amplitude and AUC of Young and YC, Pre-Post CdCl$_2$ with statistical differences between pre CdCl$_2$ and post CdCl$_2$. Related to *Figure 7*.

| Group | n | ADP Amplitude (mV) pre | ADP Amplitude (mV) post | Pre-Post p-value | AUC (mV*ms) pre | AUC (mV*ms) post | Pre-Post p-value |
|---|---|---|---|---|---|---|---|
| YN | 5 | 2.74 ± 0.79 | 0.81 ± 0.11 | 0.0806~ | 144466 ± 30557 | 47629 ± 12450 | 0.0461* |
| YC | 3 | 4.72 ± 0.74 | 0.22 ± 0.27 | 0.0067** | 323777 ± 41691 | 1014 ± 27220 | 0.0005** |

*n*, number of cells in each group; p-value, pre-post, Sidak's multiple comparisons test.

and inhibition of persistent activity impairs acquisition (*Fuster, 1972*; *Liu et al., 2014*). On the level of individual neurons, persistent firing ability as we have described here may support the trace by spanning the entire trace interval (*Constantinidis et al., 2018*) or allow the neuron to participate in an ensemble pattern of activity that allows the trace to be bridged (*Miller et al., 2018*; *Lundqvist et al., 2018*). Either way, persistent firing in LEC III is strongly suggested to support the acquisition of trace eyeblink conditioning. Blocking activity of the LEC has been shown to inhibit temporal associative learning (*Morrissey et al., 2012*; *Wilson et al., 2013*), and in the medial entorhinal cortex, inhibition of layer III activity to the CA1 region has been shown to impair trace fear conditioning (*Suh et al., 2011*). Together, these data suggest that persistent firing in LEC III is a crucial mechanism in supporting the formation of the temporal association during trace eyeblink conditioning.

Whatever the functional role of persistent firing may be in supporting trace eyeblink conditioning, what we have established is that it is a mechanism that is, at the very least, important for acquiring this temporal associative paradigm. Successful acquisition of trace eyeblink conditioning enhances persistent firing ability in both young adult and aged animals, while an aged-related decrease in persistent firing ability may lead to learning impairments in acquiring the paradigm. Along with examining the functional role of persistent firing in acquiring temporal associative learning in vivo, future studies should also examine the extent to which persistent firing is necessary. Such studies would pave the way to better understanding the neural network that underlies the acquisition of this type of complex task.

# Materials and methods

**Key resources table**

| Reagent type (species) or resource | Designation | Source or reference | Identifiers | Additional information |
|---|---|---|---|---|
| Chemical compound, drug | Carbachol | EMD Millipore | Cat#212385 | 10 µM |
| Chemical compound, drug | Cadmium Chloride (CdCl$_2$) | Millipore Sigma | Cat#10108-64-2 | 100 µM |
| Strain, strain background | F1 Hybrid F344 x Brown Norway Rat | National Institutes of Aging | RRID:SCR_007317 | |
| Software, algorithm | LabView 8.20 | National Instruments | RRID:SCR_014325 | |
| Software, algorithm | Prism 8.3.1 | GraphPad | RRID:SCR_002798 | |
| Software, algorithm | Anaconda Navigator 1.9.6 | Python | https://docs.anaconda.com/anaconda/install/ | |
| Software, algorithm | MATLAB R2014b | Mathworks | RRID:SCR_001622 | |

## Experimental model and subject details

Subjects were young adult (3–6 months old) and aged (26–32 months old) male F1 hybrid Fischer 344 x Brown Norway rats. The rats were housed in a climate-controlled vivarium on a 14:10 light: dark cycle with ad libitum access to food and water. Animal care and experimental protocols were consistent with National Institutes of Health guidelines and approved by the Northwestern University Institutional Animal Care and Use Committee.

There were six behavioral groups: Young Naive (n = 40 animals), Adult Naive (n = 30), Young Conditioned (n = 21), Young Pseudoconditioned (n = 19), Aged Unimpaired (n = 21), and Aged Impaired (n = 16). Young Conditioned, Young Pseudoconditioned, Aged Unimpaired, and Aged Impaired rats were trained on trace eyeblink conditioning as previously described (*Knuttinen et al., 2001*; *McKay et al., 2012*; *McKay et al., 2013*; *Lin et al., 2016*). Aged Unimpaired and Aged Impaired were separated into their groups based on whether they reached a 60% conditioned response learning criterion. Young rats that were to be trained on trace eyeblink conditioning were allocated to Young Conditioned and Young Pseudoconditioned groups randomly.

## Method details

### Surgery
Using sterile surgical techniques and under isoflurane inhalation anesthesia, rats were surgically implanted with a headbolt designed to deliver an electrical shock to the periorbital region of the right eye, as well as to record EMG activity from the upper right eyelid. The headbolt contained two wires to deliver an electrical shock, two wires to record EMG activity, and a final wire to serve as a ground.

### Eyeblink conditioning
Rats were allowed a minimum of 5 days to recover from surgery before training. Rats were trained over 3 days, with two sessions occurring per day (one in the morning, one in the afternoon). The first session on the first day was a habituation session to allow the rats to habituate to the training apparatus. In total, the rats received one session of habituation and five training sessions of trace eyeblink conditioning. Rats were trained in a sound-attenuating chamber allowing for free movement. Rats were trained with a tone as the conditioned stimulus (CS; 250 ms, 8 kHz, 85 dB free field) paired with an electrical shock to the periorbital region of the right eye as the unconditioned stimulus (US; 100 ms; 15–20 mA; biphasic at 60 Hz). The CS and the US were separated with a trace period of 500 ms. A single training session consisted of 30 CS-US pairings with an intertrial interval of 20 to 40 s (30 s average). EMG activity exceeding 4 SD above baseline activity occurring 200 ms before US onset was considered a conditioned response (CR). A criterion of 60% CRs was set to separate the impaired learners from the unimpaired learners. Young Pseudoconditioned rats were given 30 unpaired CS alone trials pseudo-randomly dispersed with 30 unpaired US alone trials with a 10 to 20 s intertrial interval (15 s average). Stimuli were delivered and data were acquired and analyzed using custom LabView software (https://github.com/JPow2020/Eyeblink-Conditioning, *Power, 2011*, copy archived at https://github.com/elifesciences-publications/Eyeblink-Conditioning; *McKay et al., 2013*; *Lin et al., 2016*).

### In-vitro whole cell recordings
Whole-cell current clamp recordings were made from visually identified LEC III neurons using an Axoclamp 2A amplifier, as previously described (*Matthews et al., 2009*; *McKay et al., 2013*). LEC slices were prepared from rats that had been trained on trace eyeblink conditioning 24 hr after the last training session. The experimenter was blinded to the training and learning condition of the animal. LEC slices were cut as described previously (*Tahvildari and Alonso, 2005*) using a Leica VT1000s vibratome in ice-cold artificial cerebral spinal fluid (ACSF) bubbled with 95% $O_2$-5% $CO_2$. Slices from young rats were cut using ACSF composed of the following (in mM): 124 NaCl, 26 $NaHCO_3$, 2.4 $CaCl_2$, 2.5 $KCl_2$, 1.25 $NaH_2PO_4$, 2 $MgSO_4$, and 25 D-glucose. Slices from aged rats were sliced using ACSF composed of the following (in mM): 206 sucrose, 26 $NaHCO_3$, 0.1 $CaCl_2$, 2.5 KCl, 1.25 $NaH_2PO_4$, 3 $MgSO_4$, and 15 D-glucose. Slices were incubated for ~20 min at 34°C and allowed to cool to room temperature for at least another 30 min before recording.

For whole-cell recordings, slices were placed in a submerged recording chamber and were perfused with oxygenated ACSF warmed to 32–34°C using an in-line heater and temperature controller (Warner Instruments). LEC layer III neurons were visualized using a CCD camera (Orca R2, Hamamatsu) mounted on a Leica DM LFS. High resistance seals (>1 GΩ) were obtained prior to whole cell recordings. The microscope was equipped with a long working distance 40 x (0.8 NA) water-immersion objective and infrared differential interference contrast optics. To be included in the study, cells were required to have a resting membrane potential between −50 mV and −80 mV, an $R_{input}$ of >25

**Table 17.** Passive membrane properties of LEC III pyramidal neurons, separated into YN, AN, YC, YP, AU, and AI.

| Group | RMP (mV) | $R_{input}$ (MΩ) | |
|---|---|---|---|
| YN | −72.28 ± 1.12 | 54.49 ± 1.39 | |
| AN | −70.73 ± 0.89 | 60.39 ± 2.91 | |

| Group | First AP Threshold (mV) | Last AP Threshold (mV) | p-Value |
|---|---|---|---|
| YN | -41.54 ± 0.33 | -41.63 ± 0.33 | AP Threshold, $F_{1, 379}$ = 0.6896, p = 0.4068 |
| | | | Group, $F_{1, 379}$ = 8.744, p = 0.003** |
| | | | Group x AP Threshold, $F_{1, 379}$ = 2.902, p = 0.0893~ |
| AN | -43.40 ± 0.48 | -43.15 ± 0.53 | YN vs. AN First AP Threshold, p = 0.0028** |
| | | | Last AP Threshold, p = 0.0178* |

| Group | First AP Half-Width (ms) | Last AP Half-Width (ms) | p-Value |
|---|---|---|---|
| YN | 1.97 ± 0.04 | 2.00 ± 0.03 | AP Half-Width $F_{1, 379}$ = 8.761, p = 0.0033** |
| | | | Group, $F_{1, 379}$ = 2.667, p = 0.1033 |
| | | | Group x AP Half-Width $F_{1, 379}$ = 3.120, p = 0.0781~ |
| AN | 1.83 ± 0.04 | 1.95 ± 0.04 | YN vs AN First AP Half-Width, p = 0.0523~ |
| | | | Last AP Half-Width, p = 0.7060 |

| Group | First AP Amplitude (mV) | Last AP Amplitude (mV) | p-Value |
|---|---|---|---|
| YN | 83.61 ± 0.44 | 80.45 ± 0.43 | Two-Way ANOVA, AP Amplitude $F_{1, 379}$ = 521.7, p < 0.0001*** |
| | | | Group, $F_{1, 379}$ = 0.001639, p = 0.9677 |
| | | | Group x AP Amplitude $F_{1, 379}$ = 0.3927, p = 0.5313 |
| AN | 83.49 ± 0.64 | 80.50 ± 0.73 | YN vs AN First AP Amplitude, p = 0.9862 |
| | | | Last AP Amplitude, p = 0.9970 |

| Group | First AP dV/dt max (v/s) | Last AP dV/dt max (v/s) | p-Value |
|---|---|---|---|
| YN | 199.3 ± 3.0 | 175.4 ± 2.9 | dV/dt Max $F_{1, 379}$ = 385.7, p < 0.0001*** |
| | | | Group, $F_{1, 379}$ = 0.4711, p = 0.4929 |
| | | | Group x dV/dt Max $F_{1, 379}$ = 0.1451, p = 0.7034 |
| AN | 202.3 ± 4.2 | 179.3 ± 4.6 | YN vs AN First dV/dt Max, p = 0.8083 |
| | | | Last dV/dt Max, p = 0.6975 |

| Group | RMP (mV) | $R_{input}$ (MΩ) | |
|---|---|---|---|
| YC | -75.35 ± 0.71 | 55.74 ± 2.53 | |
| YP | -74.37 ± 0.86 | 57.66 ± 3.70 | |
| AU | -72.71 ± 0.76 | 59.87 ± 2.27 | |
| AI | -70.24 ± 1.40 | 59.06 ± 2.95 | |

| Group | First AP Threshold (mV) | Last AP Threshold (mV) | p-Value |
|---|---|---|---|
| YC | -37.33 ± 0.40 | -37.39 ± 0.47 | AP Threshold, $F_{1, 643}$ = 15.71, p < 0.0001*** |

*Table 17 continued on next page*

*Table 17 continued*

| Group | RMP (mV) | R$_{input}$ (MΩ) | | | |
|---|---|---|---|---|---|
| YP | -40.70 ± 0.36 | -41.46 ± 0.35 | Group,<br>F$_{3, 643}$ = 20.74, p < 0.0001*** | | |
| | | | Group x AP Threshold,<br>F$_{3, 643}$ = 1.826, p = 0.1412 | | |
| AU | -40.80 ± 0.35 | -41.10 ± 0.34 | Comparison | First AP | Last AP |
| | | | YC vs YP | p < 0.0001*** | p < 0.0001*** |
| | | | YC vs AU | p < 0.0001*** | p < 0.0001*** |
| | | | YC vs AI | p < 0.0001*** | p < 0.0001*** |
| AI | -40.69 ± 0.39 | -41.17 ± 0.32 | YP vs AU | p = 0.9974 | p = 0.9062 |
| | | | YP vs AI | p > 0.9999 | p = 0.9488 |
| | | | AU vs AI | p = 0.9947 | p = 0.9990 |

| Group | First AP Half-Width (ms) | Last AP Half-Width (ms) | p-Value | | |
|---|---|---|---|---|---|
| YC | 2.23 ± 0.06 | 2.32 ± 0.09 | AP Half-Width,<br>F$_{1, 643}$ = 4.928e-005, p = 0.9944 | | |
| YP | 1.96 ± 0.05 | 1.94 ± 0.04 | Group,<br>F$_{3, 643}$ = 28.46, p < 0.0001*** | | |
| | | | Group x Half-Width<br>F$_{3, 643}$ = 1.650, p = 0.1767 | | |
| AU | 1.82 ± 0.04 | 1.80 ± 0.04 | Comparison | First AP | Last AP |
| | | | YC vs YP | p = 0.0007** | p < 0.0001*** |
| | | | YC vs AU | p < 0.0001*** | p < 0.0001*** |
| | | | YC vs AI | p < 0.0001*** | p < 0.0001*** |
| | | | YP vs AU | p = 0.1239 | p = 0.1198 |
| AI | 1.82 ± 0.04 | 1.77 ± 0.03 | YP vs AI | p = 0.1361 | p = 0.0494* |
| | | | AU vs AI | p > 0.9999 | p = 0.9682 |

| Group | First AP Amplitude (mV) | Last AP Amplitude (mV) | p-Value | | |
|---|---|---|---|---|---|
| YC | 80.02 ± 1.11 | 76.10 ± 1.07 | AP Amplitude,<br>F$_{1, 643}$ = 286.5, p < 0.0001 | | |
| YP | 81.80 ± 0.88 | 78.92 ± 0.82 | Group,<br>F$_{3, 643}$ = 3.996, p = 0.0078 | | |
| | | | Group x AP Amplitude,<br>F$_{3, 643}$ = 5.083, p = 0.0017 | | |
| AU | 83.10 ± 0.66 | 80.27 ± 0.52 | Comparisons | First AP | Last AP |
| | | | YC vs YP | p = 0.5087 | p = 0.1274 |
| | | | YC vs AU | p = 0.0404* | p = 0.0019** |
| | | | YC vs AI | p = 0.9990 | p = 0.2916 |
| AI | 80.18 ± 0.83 | 78.19 ± 0.75 | YP vs AU | p = 0.6715 | p = 0.6391 |
| | | | YP vs AI | p = 0.5082 | p = 0.9233 |
| | | | AU vs AI | p = 0.0237* | p = 0.1762 |

| Group | First AP dV/dt max (v/s) | Last AP dV/dt max (v/s) | p-Value | | |
|---|---|---|---|---|---|
| YC | 178.0 ± 5.1 | 151.1 ± 4.6 | AP dV/dt max,<br>F$_{1, 643}$ = 15.71, p < 0.0001 | | |
| YP | 183.5 ± 5.3 | 163.8 ± 4.6 | Group,<br>F$_{3, 643}$ = 20.74, p < 0.0001 | | |
| | | | Group x AP dV/dt max,<br>F$_{3, 643}$ = 1.826, p = 0.1412 | | |
| AU | 191.0 ± 3.7 | 172.4 ± 2.9 | Comparison | First AP | Last AP |
| | | | YC vs YP | p = 0.8306 | p = 0.2052 |

*Table 17 continued on next page*

*Table 17 continued*

| Group | RMP (mV) | R<sub>input</sub> (MΩ) | | | |
|-------|----------|------------------------|------------|------------|------------|
| AI | 176.7 ± 4.2 | 162.0 ± 3.2 | YC vs AU | p = 0.1219 | p = 0.0017** |
| | | | YC vs AI | p = 0.9966 | p = 0.2653 |
| | | | YP vs AU | p = 0.5757 | p = 0.4517 |
| | | | YP vs AI | p = 0.6563 | p = 0.9888 |
| | | | AU vs AI | p = 0.0313* | p = 0.1836 |

All statistical analyses were first performed with a Two-Way ANOVA. *Post hoc* multiple comparisons tests performed on behaviorally naïve Young and Aged data were done with Sidak's tests. Multiple comparisons performed on YC, YP, AU, and AI data were done with Tukey's tests.

MΩ that remained stable throughout the recording, and an action potential height of at least 70 mV above the holding potential. $R_{input}$ was calculated by a linear fit of the steady state voltage response (average membrane potential 700–900 ms of the 1 s current step) vs current steps (−0.3 to 0 nA, 0.05 nA increments). Action potential properties (i.e. threshold, half-width, amplitude, dV/dt max) were measured from the first action potential evoked by persistent firing and the last action potential recorded in the sweep (*Table 17*). Recordings were acquired and analyzed using pClamp v10 and digitized using a Digidata 1440A analog-to-digital converter. Custom routines using Python were written to analyze $R_{input}$, and action potential threshold, half-width, amplitude, and dV/dt.

## Persistent firing recordings

All persistent firing recordings were performed using patch electrodes containing (in mM): 120 K-Gluconate, 10 HEPES, 0.2 EGTA, 20 KCl, 2 $MgCl_2$, 7 PhCreat di(tris), 4 Na-ATP, 0.3 GTP, and 0.1% biocytin (pH adjusted to 7.3 with KOH, 280 ± 5 mOsm), as described in *Yoshida et al., 2008*; *Yoshida and Hasselmo, 2009*. Intrinsic recordings of excitability (i.e. postburst AHP) were performed using patch electrodes (5–8 MΩ) containing (in mM): 120 $KMeSO_4$, 10 KCl, 10 Hepes, 4 $Mg_2ATP$, 0.4 NaGTP, 10 $Na_2$phosphocreatine, 0.5% neurobiotin, pH adjusted to 7.3 with KOH, 280 ± 5 mOsm. The experimenters were not able to elicit persistent firing using solutions containing 120 $KMeSO_4$.

Persistent firing was evoked by bath applying 10 µM carbachol (*Tahvildari et al., 2007*; *Tahvildari et al., 2008*; *Yoshida et al., 2008*; *Yoshida and Hasselmo, 2009*). Persistent firing was evoked with three different types of protocols: 1) with a 2 s long training stimulus of 100 pA, 150 pA, and 200 pA amplitudes while the cell's membrane potential was held at 2 mV below spontaneous firing threshold and 2) with a 2 s long train of 2 ms, 2 nA current injection pulses at 20 Hz while the cell was held at 2 mv below spontaneous firing threshold and 3) with a 250 ms long train of 2 ms, 2 nA current injection pulses at 20 Hz while the cell's membrane potential was held at 2 mV and 5 mV more hyperpolarized than its spontaneous firing threshold. Each cell's unique spontaneous firing threshold was determined before the protocols were evoked and measured at intervals throughout the recording session to ensure threshold had not drifted. Persistent firing activity was measured during 20 s long sweeps. Each protocol was evoked three times per cell and the recorded activity was averaged across the three sweeps. There was an average of 1 min interval between each sweep. An incidence of persistent firing was considered if the firing activity occurred within 10 s after the end of the current injection. The average group probability of persistent firing for each protocol was determined as the averaged probability of all cells, based on the proportion of sweeps each cell persistently fired.

Measurements of the afterdepolarization (ADP) and plateau potential (PP) were made while the cell's membrane potential was held at 10 mV more hyperpolarized than its spontaneous firing threshold to ensure that persistent firing would not be evoked. The ADP and PP was then evoked with a 250 ms long train of 2 ms, 2 nA current injection pulses at 20 Hz. Firing rate analyses were performed only from neurons that were able to persistently fire at least once out of the three sweeps (i.e. if a neuron was at 0% firing ability, that neuron was not included in the frequency analysis.) Firing rate was measured in 1 s bins after the offset of the training stimulus. In bins in which there were no action potentials, a value of 0 was assigned to the bin. The mean firing rate of the neuron was determined by averaging the firing rate across the three sweeps. If the neuron was able to persistently

fire for one, but not for all three sweeps, a value of 0 was assigned to all bins in the sweeps in which the neuron did not persistently fire. The group firing rate was determined by averaging the mean firing rate of each neuron in that group.

For experiments involving $CdCl_2$ a solution of 100 μM $CdCl_2$ and 10 μM carbachol in ACSF containing 124 NaCl, 26 $NaHCO_3$, 2.4 $CaCl_2$, 2.5 KCl, 2 $MgSO_4$, and 25 D-glucose was prepared and perfused through the recording chamber for at least 10 min before recording.

Custom Python codes were written to analyze the rate of persistent firing, latency to onset, ADP size, and the AUC of the ADP and PP.

## Postburst AHP recordings

For intrinsic excitability recordings, neurons were held at −69 mV (−67 to −71 mV), near the cell's normal resting membrane potential. The postburst AHP was evoked by a train of 15 action potentials that were induced by 15 (2 ms, 2 nA) current injection pulses at 50 Hz. The postburst AHP was evoked five times per cell with a 30 s interval between each sweep and the activity averaged across all five sweeps. The medium AHP was measured as the difference between the holding potential and the negative-going peak of the membrane potential after the offset of the last current step. The slow AHP was measured as the difference between holding potential and the membrane potential 1 s after the offset of the last current step. A custom MATLAB routine was used to analyze the postburst AHP.

## Statistical analysis

All statistical analyses were performed using GraphPad Prism. Descriptive statistics are reported as mean ± SEM unless otherwise noted. All experiments and statistical analyses were planned a priori. All ANOVAs and Kruskal-Wallis tests are reported in the body of the text, while t-tests, Mann-Whitney tests, and multiple comparisons tests are reported in the tables.

All analyses of the probability of firing were performed using non-parametric statistical tests, given the nature of the data (i.e. Mann-Whitney, Kruskal-Wallis, Dunn's multiple comparisons). Other data were analyzed using parametric statistical tests (t-tests, ANOVA, Tukey's, Sidak) unless they did not meet the assumptions of parametric statistical tests. Statistical significance was defined as $p < 0.05$.

All statistical analyses were first performed with a Two-Way ANOVA. *Post hoc* multiple comparisons tests performed on behaviorally naïve Young and Aged data were done with Sidak's tests. Multiple comparisons performed on YC, YP, AU, and AI data were done with Tukey's tests.

## Acknowledgements

We thank Dr. Motoharu Yoshida for advice regarding evoking persistent firing in entorhinal neurons. We thank Michael McCarthy and Natalia Vilcek for training the rats on trace eyeblink conditioning and analyzing the behavioral data. We thank Dr. John Power for writing the eyeblink conditioning LabView software that was used to train the animals and analyze the behavioral data. We thank Dr. Shoai Hattori for writing the Matlab code to analyze the postburst AHP data. This work was supported by NIH Grant R37AG008796 (to JFD), RF1AG017139 (to JFD) and F31AG055331 (to CL).

## Additional information

### Funding

| Funder | Grant reference number | Author |
| --- | --- | --- |
| National Institutes of Health | AG008796 | John F Disterhoft |
| National Institutes of Health | AG017139 | John F Disterhoft |
| National Institutes of Health | AG55331 | Carmen Lin |

The funders had no role in study design, data collection and interpretation, or the decision to submit the work for publication.

## Author contributions
Carmen Lin, Conceptualization, Data curation, Formal analysis, Investigation, Visualization, Methodology, Writing - original draft, Project administration, Writing - review and editing; Venus N Sherathiya, Software; M Matthew Oh, John F Disterhoft, Conceptualization, Supervision, Writing - review and editing

## Author ORCIDs
Carmen Lin (ID) https://orcid.org/0000-0003-2329-2782
Venus N Sherathiya (ID) https://orcid.org/0000-0003-0143-9771
M Matthew Oh (ID) https://orcid.org/0000-0002-6702-5785

## Ethics
Animal experimentation: The study was performed in strict accordance with the protocols (IS00002081) approved by the Northwestern University Institutional Animal Care and Use Committee, consistent with National Institutes of Health guidelines. All surgery was performed under isoflurane inhalation anesthesia and all effort was made to minimize suffering.

## Decision letter and Author response
Decision letter https://doi.org/10.7554/eLife.56816.sa1
Author response https://doi.org/10.7554/eLife.56816.sa2

# Additional files

## Data availability
All data and code generated for this study have been deposited in Dryad and is entitled "Persistent Firing in LEC III Neurons is Differentially Modulated by Learning and Aging." The eyeblink conditioning LabView software has been deposited in GitHub and is available via this link: https://github.com/JPow2020/Eyeblink-Conditioning (copy archived at https://github.com/elifesciences-publications/Eyeblink-Conditioning).

The following dataset was generated:

| Author(s) | Year | Dataset title | Dataset URL | Database and Identifier |
|---|---|---|---|---|
| Lin C, Sherathiya VN, Oh MM, Disterhoft JF | 2020 | Persistent Firing in LEC III Neurons is Differentially Modulated by Learning and Aging | http://dx.doi.org/10.5061/dryad.tqjq2bvvr | Dryad Digital Repository, 10.5061/dryad.tqjq2bvvr |

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
