## [Decision Letter]

**Acceptance summary:**

This paper provides new insight linking changes in the biophysical properties of single neurons with age-related decline in learning. Using eye-blink conditioning in young and old rats, the authors demonstrate that increasing age decreases learning in this condition. They then go on to show how this change in learning correlates with an age-related decline in persistent firing (i.e., the ability of neurons to maintain a high firing rate even after the removal of a stimulus) in the entorhinal cortex, which has been proposed to serve a key role in working and associative memory processes. Consequently, this work has the potential to provide new insight on the circuit mechanisms of memory within the entorhinal-hippocampal network and how entorhinal dysfunction contributes to memory deficits in aging.

**Decision letter after peer review:**

Thank you for submitting your article "Persistent firing in LEC III neurons is differentially modulated by learning and aging" for consideration by *eLife*. Your article has been reviewed by three peer reviewers, and the evaluation has been overseen by a Reviewing Editor and Laura Colgin as the Senior Editor. The following individuals involved in review of your submission have agreed to reveal their identity: James Alexander Ainge (Reviewer #1); Kei Igarashi (Reviewer #3).

The reviewers have discussed the reviews with one another and the Reviewing Editor has drafted this decision to help you prepare a revised submission.

Summary:

The paper presents a series of studies examining the effects of learning and aging on persistent firing of layer 3 neurons within the lateral entorhinal cortex (LEC). The studies examine eye-blink conditioning in young and old rats and show interesting opposing effects such that increasing age has the opposite effect of learning. The studies are well-designed and present an interesting mechanism for how changes in persistent firing may underlie associative memory within the LEC. Moreover, the paper goes on to show how this mechanism is affected in aging. Consequently, this work has the potential to provide new insight on the circuit mechanisms of memory within the entorhinal-hippocampal network and how entorhinal dysfunction contributes to memory deficits in aging.

Essential revisions:

The reviewers felt there were two major avenues or revision that are needed for publication. The first is a series of analyses that should be feasible to conduct on the current data set. The second is re-writing portions of the paper to improve clarity, flow and the communication of the results.

Analyses:

1) Reviewers were concerned there could be other behavioral differences between young and old animals that could be measured using the current data set (levels of locomotion or anxiety, for example).

2) The authors should provide justification of the 60% conditioning threshold. It would be informative to know if the distribution of conditioning scores is bimodal or unimodal. A bimodal distribution would suggest that the rats can be split into two groups whose responses are qualitatively as well as quantitatively different; learners vs. non-learners. A unimodal distribution would call into question the validity of splitting the group in 2. Collapsing animals into 2 groups loses a lot of the variance in the data. A more interesting approach would to examine the exact relationship between electrophysiological measures and level of conditioning using some form of regression or correlation. This would capture much more of the variance in the data and remove the arbitrary 60% cut-off.

3) The authors should provide and criteria for splitting the data into slow and medium AHP.

4) The authors should include text indicating the implications of the slower time to reach peak firing rate Figure 5?

Writing and Discussion:

1) The authors found the organization of the manuscript difficult to follow. For example, Figure 1 discusses the mechanism underlying persistent firing while Figure 2 discusses the properties of persistent firing but it may be clearer to present the generate properties of persistent firing before mechanism. A suggested order was: 1) Comparison of persistent firing between naïve young vs. naïve old, 2) Effect of learning in young rats (comparing naïve, learned and unlearned young rats), 3) Effect of learning in old rats(comparing naïve, learned and unlearned old rats), 4) Assessment of AHP and ADP, 5) Ca^2+^-dependence of AHP. For the Discussion, a suggested order was: 1) Summary of the data, 2) The potential mechanisms of persistent firing, 3)Mechanistic discussion of aging and learning on the persistent firing, 4) Potential role in vivo.

2) Please expand upon the following points in the Discussion section: 1) Similar age- and learning-dependent effects on intrinsic excitability were reported in several regions, including the hippocampus (e.g., Moyer et al., 1996, 2000; Matthews et al., 2009) and medial prefrontal cortex (e.g., Kaczorowski et al., 2012). Therefore, intrinsic excitability might change everywhere with aging, which raises a question as to what unique contribution the reported changes in the LEC would make to learning deficits in aged rats. 2) The authors state that "The ability of AU animals to achieve the same level of learning behaviorally must be the result of compensatory mechanisms other than those mediated by persistent firing in LEC III neurons." Please expand this point by listing some brain regions that may compensate for the LEC dysfunction. 3) The relationship between the ADP and AHP is unclear and they may be interdependent (small AHP may produce a large ADP). Please discuss the relationship between them.

3) There were several places where some re-writing would improve clarity. There are multiple long sentences that can be shortened, the definition of ADP and PP is unclear and the definition of mAHP and sAHP is unclear.

4) The reviewers indicated that some portions of the text were overstatements of the previous literature. For example, the authors state that "Despite evidence that persistent firing in LEC III supports temporal associative learning…", however, this has not been established. The authors cite Suter et al. paper, but in this paper, they only observed firing of LEC neurons during CS which slightly prolonged into the delay phase and decreased back to *baseline*. The paper itself did not conclude/discuss about the persistent firing. The authors should indicate that the role of LEC persistent firing in learning is still unclear.

---

## [Author Response]

Essential revisions:The reviewers felt there were two major avenues or revision that are needed for publication. The first is a series of analyses that should be feasible to conduct on the current data set. The second is re-writing portions of the paper to improve clarity, flow and the communication of the results.Analyses:1) Reviewers were concerned there could be other behavioral differences between young and old animals that could be measured using the current data set (levels of locomotion or anxiety, for example).

We thank the reviewers for raising this point. However, we did not measure or record other behavioral differences such as levels of locomotion or anxiety from our animals. Although there is a possibility that there may be a difference, we did not note any differences between our young and aged animals in this particular experiment nor in any of our previous experiments involving young and aged animals performing trace eyeblink conditioning (Knuttinen et al., 2001; Matthews et al., 2009).

2) The authors should provide justification of the 60% conditioning threshold. It would be informative to know if the distribution of conditioning scores is bimodal or unimodal. A bimodal distribution would suggest that the rats can be split into two groups whose responses are qualitatively as well as quantitatively different; learners vs. non-learners. A unimodal distribution would call into question the validity of splitting the group in 2. Collapsing animals into 2 groups loses a lot of the variance in the data. A more interesting approach would to examine the exact relationship between electrophysiological measures and level of conditioning using some form of regression or correlation. This would capture much more of the variance in the data and remove the arbitrary 60% cut-off.

We thank the reviewers for this point regarding the separation of the aging group of animals into learners vs. non-learners. In response to this comment, we included the frequency distribution of the %CR on the final session of trace eyeblink conditioning (session 5) for the aged animals (Figure 2). The distribution shows a split in behavior at the 55% CR mark, which informed our decision to separate the aged animals into Aged Unimpaired and Aged Impaired based on a 60% CR criterion. We included text in the subsection “Successful Learning Enhances Persistent Firing Probability, and Learning Impairments are Associated with Lower Firing Probability” to explain the decision for a 60% CR criterion.

We also believe, however, that a regression analysis would allow us to gain some insight into the relationship between behavior and certain electrophysiological measures. To that end, we ran Pearson’s correlation tests on the % CR from session 5 of trace eyeblink conditioning against persistent firing rate, postburst AHP amplitude, and ADP amplitude. We found that there was a positive correlation between behavior and these electrophysiological measures for the aged group of animals that underwent trace eyeblink conditioning (AU and AI). However, the correlation was not present for the young group of animals that underwent trace eyeblink conditioning (YC). We have included these correlations as supplementary figures (Figure 4—figure supplement 2, Figure 5—figure supplement 2, Figure 6—figure supplement 2) and included text in the subsections “Aging Decreases Persistent Firing Rate while Learning Increases Firing Rate”, “The Postburst AHP is altered with Aging and Learning in LEC III Pyramidal Neurons” and “Persistent Firing Properties are Dependent on ADP Size”.

3) The authors should provide and criteria for splitting the data into slow and medium AHP.

We have included text explaining the criteria for splitting the postburst AHP into the medium and slow AHP (subsection “The Postburst AHP is altered with Aging and Learning in LEC III Pyramidal Neurons”).

4) The authors should include text indicating the implications of the slower time to reach peak firing rate Figure 5?

We have included text in the Discussion section indicating the implications of the slower time to reach peak firing rate amongst the aged group of animals (Discussion, sixth paragraph).

Writing and Discussion:1) The authors found the organization of the manuscript difficult to follow. For example, Figure 1 discusses the mechanism underlying persistent firing while Figure 2 discusses the properties of persistent firing but it may be clearer to present the generate properties of persistent firing before mechanism. A suggested order was: 1) Comparison of persistent firing between naïve young vs. naïve old, 2) Effect of learning in young rats (comparing naïve, learned and unlearned young rats), 3) Effect of learning in old rats(comparing naïve, learned and unlearned old rats), 4) Assessment of AHP and ADP, 5) Ca^2+^-dependence of AHP. For the Discussion, a suggested order was: 1) Summary of the data, 2) The potential mechanisms of persistent firing, 3)Mechanistic discussion of aging and learning on the persistent firing, 4) Potential role in vivo.

We thank the reviewers for bringing the organization of the manuscript to our attention. We understand the confusion regarding placing the postburst AHP, a mechanism underlying changes in persistent firing in Figure 1 before discussing persistent firing properties in Figure 2. We agree that placing the postburst AHP section after the comparisons of persistent firing properties in young and aged rats is better for organization. However, we wished to preserve the group comparisons that we had already done (i.e. young naïve vs. aged naïve and young conditioned vs. young pseudo vs. aged unimpaired vs. aged impaired). We believe the comparisons between young conditioned and aged unimpaired offered insight into the effects of aging-related changes to persistent firing on learning (i.e. that aged unimpaired and young conditioned reached the same level behaviorally but not in terms of persistent firing).

However, we do agree that there is insight to be gained from comparisons of learning on all young rats (i.e. young naïve vs. young conditioned vs. young pseudo) and on all aged rats (i.e. aged naïve vs. aged unimpaired vs. aged impaired). To that end, we have included supplementary figures (Figure 2—figure supplement 1; Figure 4—figure supplement 1; Figure 5—figure supplement 1; Figure 6—figure supplement 1) which shows the differences in persistent firing probability, persistent firing rate, postburst AHP amplitude, and ADP size amongst all young rats and all aged rats. We have also included text in the subsections “Successful Learning Enhances Persistent Firing Probability, and Learning Impairments are Associated with Lower Firing Probability”, “Aging Decreases Persistent Firing Rate while Learning Increases Firing Rate”, “The Postburst AHP is altered with Aging and Learning in LEC III Pyramidal Neurons” and “Persistent Firing Properties are Dependent on ADP Size”.

We have reorganized the manuscript as follows: 1) Comparison of persistent firing between young and aged naïve, 2) Comparison of persistent firing between young conditioned, young pseudoconditioned, aged unimpaired, and aged impaired, 3) Assessment of persistent firing rate and onset latency, 4) Assessment of postburst AHP, 5) Assessment of ADP, and 6) Ca^2+^-dependence of the ADP. We have also reorganized the Discussion as the reviewers suggested.

2) Please expand upon the following points in the Discussion section: 1) Similar age- and learning-dependent effects on intrinsic excitability were reported in several regions, including the hippocampus (e.g., Moyer et al., 1996, 2000; Matthews et al., 2009) and medial prefrontal cortex (e.g., Kaczorowski et al., 2012). Therefore, intrinsic excitability might change everywhere with aging, which raises a question as to what unique contribution the reported changes in the LEC would make to learning deficits in aged rats. 2) The authors state that "The ability of AU animals to achieve the same level of learning behaviorally must be the result of compensatory mechanisms other than those mediated by persistent firing in LEC III neurons." Please expand this point by listing some brain regions that may compensate for the LEC dysfunction. 3) The relationship between the ADP and AHP is unclear and they may be interdependent (small AHP may produce a large ADP). Please discuss the relationship between them.

We chose to address the first two points together regarding the unique contributions of aging-related changes to intrinsic excitability in LEC III and the brain regions that may compensate for LEC III in aging. To address these points, we have included text in the Discussion.

We have also included text in the Discussion to address the relationship between the AHP and the ADP.

3) There were several places where some re-writing would improve clarity. There are multiple long sentences that can be shortened, the definition of ADP and PP is unclear and the definition of mAHP and sAHP is unclear.

We have gone through the manuscript and shortened some of the longer sentences. We have also included definitions of the ADP and PP (subsection “Persistent Firing Properties are Dependent on ADP Size”) and the mAHP and the sAHP (subsection “The Postburst AHP is altered with Aging and Learning in LEC III Pyramidal Neurons”).

4) The reviewers indicated that some portions of the text were overstatements of the previous literature. For example, the authors state that "Despite evidence that persistent firing in LEC III supports temporal associative learning…", however, this has not been established. The authors cite Suter et al. paper, but in this paper, they only observed firing of LEC neurons during CS which slightly prolonged into the delay phase and decreased back to baseline. The paper itself did not conclude/discuss about the persistent firing. The authors should indicate that the role of LEC persistent firing in learning is still unclear.

We apologize for the overstatements of the role of persistent firing in learning. In response to these comments, we have edited the manuscript to ensure that the role of LEC III persistent firing in learning is not overstated and removed the mention of the Suter et al., 2019 paper.